# 3CIL: CAUSALITY-INSPIRED CONTRASTIVE CONDITIONAL IMITATION LEARNING FOR AUTONOMOUS DRIVING

## ABSTRACT

Imitation learning (IL) aims to recover an expert's strategy by performing supervised learning on the demonstration datasets. Incorporating IL in safety-crucial tasks like autonomous driving is promising as it requires less interaction with the actual environment than reinforcement learning approaches. However, the robustness of IL methods is often questioned, as phenomena like causal confusion occur frequently and hinder it from practical use. In this paper, we conduct causal reasoning to investigate the crucial requirements for the ideal imitation generalization performance. With insights derived from modeled causalities, we propose causality-inspired contrastive conditional imitation learning (3CIL), a conditional imitation learning method equipped with contrastive learning and action residual prediction tasks, regularizing the imitator in causal and anti-causal directions. To mitigate the divergence with experts in unfamiliar scenarios, 3CIL introduces a sample-weighting term that transforms the prediction error into an emphasis on critical samples. Extensive experiments in the CARLA simulator show the proposed method significantly improves the driving capabilities of models.

## 1 INTRODUCTION

Known as learning from demonstrations, imitation learning (IL) has attracted attention for its capability of replicating the behavior of experts in some tasks given only experts' demonstrations available. In many real-world applications, IL is widely used, because the desired behavior patterns are hard to construct, and the requirements for a competent model are difficult to summarize into a reward function or optimize objective. IL approaches can be divided into two categories, namely behavior cloning (BC) and inverse reinforcement learning (IRL). BC is one of the most prominent IL algorithms, which transforms the procedure of imitating experts' strategy into a simple supervised learning problem. For those tasks with strict safety concerns or expensive trial costs like autonomous driving, BC is preferred as provides an effective solution with no requirement on interactions with the real environment during training.

However, the performance of BC methods is often questionable, especially in complex environments. Many approaches (de Haan et al., 2019; Wen et al., 2020; Codevilla et al., 2019; Wen et al., 2022; Ortega et al., 2021) have investigated the factors that led to the problematic decision-making pattern of the imitator. What is common in these analyses is that they attribute the train-test performance gap to causal confusion: the reliance of the imitator on spurious correlations or shortcuts, instead of causal relations. Due to the lack of causal principles, models are prone to use features that are spurious correlated to the expert's actions, as relying on these correlations only needs fewer parameter updates to obtain a stable and low loss in the training phase. Moreover, the potential perception mismatch between the expert and the imitator may further impose the imitator's reliance on spurious correlations. Nevertheless, these correlations or shortcuts may hold only when the target environment has the same distribution as the distribution demonstrations sampled from. Therefore, imitators that are tempted by these shortcuts, perform poorly when testing them in new environments.

To alleviate the above problems of BC in visually complex tasks like autonomous driving, we incorporate the idea of causal reasoning to assist imitation. While typical techniques of causal reasoning

can not directly be deployed in these tasks with high-dimensional observations and partial visibility, modeling causal relations among concepts in the driving task can still provide indications for learning a robust imitator. By investigating the correspondence between causal relations and the imitator's behaviors, we identify crucial traits that a robust imitator must have. With supervisions from both causal direction and anti-causal direction, the latent state inferred by the imitator is urged to produce stable causal effects on its descendant node.

In this paper, we consider imitating experts' behavior to achieve autonomous driving, under the setting of conditional imitation learning (CIL) (Codevilla et al., 2018). Equipped with causality, we propose Causality-inspired Contrastive Conditional Imitation Learning (3CIL), an imitation learning method that incorporates contrastive learning and residual prediction tasks for better generalization. Our contributions are:

- Based on causal reasoning about the behavior cloning process in the driving task, we identify crucial traits that a robust imitator must have. By incorporating contrastive learning and action residual prediction objectives into imitation learning, we enhance the imitator's representation's robustness through influence from causal and anti-causal directions.

- We propose a sample-weighting process to emphasize scenarios that cause high divergences between the expert and imitator, guide the imitator to adapt to diverse situations.

- We conduct extensive experiments on the CARLA simulator (Dosovitskiy et al., 2017) to demonstrate the effectiveness of the proposed 3CIL approach and the relations between causal insights and actual performance.

## 2 PRELIMINARIES AND DEFINITIONS

### 2.1 SPURIOUS CORRELATIONS IN BEHAVIOR CLONING

Compared to online reinforcement learning (RL) or IRL methods, agents trained with BC are more vulnerable to spurious correlations in data. As the imitator cannot interact with the target environment in the training phase, it can not test or validate its learned pattern but only count on offline evaluation metrics (e.g. frame-wise Mean Squared Error in steer angle prediction). Such phenomenon is known in the literature as causal confusion (de Haan et al., 2019): BC agents lack explicit causal understanding of their tasks.

Causal confusion becomes more evident in complex tasks like autonomous driving. As special cases of causal confusion, inertia problem (Codevilla et al., 2019) and copycat problem (Wen et al., 2020) are proposed to describe the strong reliance of an imitator's policy on the expert's previous actions, even when such actions are no explicitly provided as input.

Figure 1a illustrates the decision process of imitators suffering from inertia and copycat problems. $o_{t-3:t}$ and $o_{j:j+1}$ denote the observations recorded in successive frames, $a_{acc,i}$ is the expert's acceleration command in current frame $i$ (i.e., $a_{acc,i} > 0$ means speed up, $a_{acc,i} < 0$ represents slow down), $v_{speed,i}$ is the speed in current frame. As discussed by (Codevilla et al., 2019), low speed often comes with negative acceleration in demonstrations, such a strong correlation tempts the inertia imitator to build a pattern: low speed causes braking. However, such a correlation exists only because the expert braked in previous frames. Moving a step forward, a copycat imitator seized that the variation in speed in previous frames provides clues of the expert's former action, and it turns to replicate the previous action to achieve low prediction error. However, this shortcut is also misleading, as in deployment time, the copycat imitator repeats its previous prediction.

Researchers (Cultrera et al., 2023; Guo et al., 2024; Seo et al., 2023; Wen et al., 2021; Samsami et al., 2021; Tien et al., 2022) have attributed such reliance on spurious correlations as (1) **The complexity of the task itself:** tasks like driving have sophisticated kinematics, diverse scenarios, continuous action space and numerous environmental parameters. (2) **The partial observability of state:** typical IL approaches to achieve driving (Hu et al., 2022a; Chuang et al., 2022) often design the expert to have access to the ground truth state (i.e., pre-trained RL agent with BEV observation or scripted expert with ground truth information), while the imitator can only receive the visual observation and measurement vector of ego vehicle. (3) **The lack of explicit causal model:** an imitator can not distinguish the causal and non-causal policies with similar offline evaluation performance, without

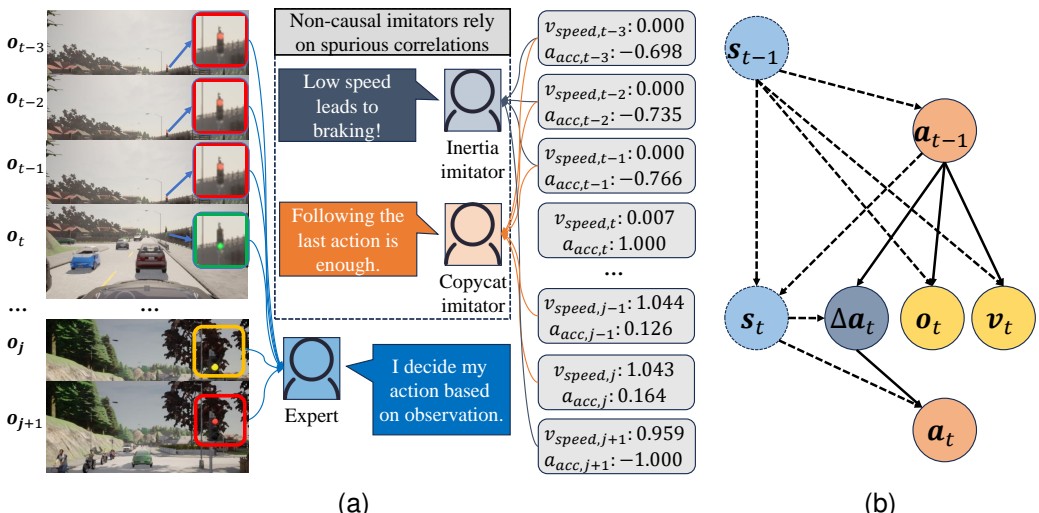

(a)                                                             (b)

Figure 1: (a): An illustration of different decision-making patterns. In timestep $t$, a 3-tuple $(\boldsymbol{o}_t, \boldsymbol{v}_t, \boldsymbol{a}_t)$ is recorded as expert demonstration. $\boldsymbol{o}_t$ denotes the image captured by the front camera, $v_{speed,t}$ is measured speed of ego vehicle, and $a_{acc,t}$ is the expert's command in acceleration ($< 0$ means braking, $> 0$ means accelerate). (b): A causal diagram of the data-generating process within two timesteps $t-1, t$ in driving tasks, subscripts representing the timestep. Dashed nodes and edges denote the variables and interactions an imitator cannot observe: as modeled in Section 2.2, the mismatched observation forms impose difficulty in recovering expert policy. $\Delta\boldsymbol{a}_t$ is the difference between previous expert action $\boldsymbol{a}_{t-1}$ and current expert action $\boldsymbol{a}_t$. $\boldsymbol{v}_t$ denotes the measurement vector that comes with the image observation $\boldsymbol{o}_t$.

prior knowledge of the causal model of the task. (4) **The evident and consistent spurious correlations:** a mapping between spurious features and the expert's previous action is easily learned and rarely violated, while differences between successive actions are usually minor. With all the factors above, an imitator prefers to infer previous actions from spurious correlated features as its prediction, instead of struggling in the large network parameter search space.

Appendix A.1 provides an introduction to works in related areas. Although methods have proposed to bring causality into fields of IL and autonomous driving, most of their approaches either focus on theoretical analysis(Howard & Kunze, 2024; Ruan et al., 2022; Ruan & Di, 2022; Kumor et al., 2021; Swamy et al., 2022b), interpreting and evaluating agents' behavior (Maier et al., 2024; Li et al., 2024; Atakishiyev et al., 2023; Jacob et al., 2022; Hart & Knoll, 2020), operating in relatively simple settings (Guo et al., 2024; Samsami et al., 2021), or designed for certain sub-tasks (Cheng et al., 2024; Hu et al., 2022b; Tang et al., 2022) instead of end-to-end driving. In contrast, we aim to use causality to assist the imitator in visually complex end-to-end driving tasks.

## 2.2 DEFINITIONS

In this paper, we consider the imitation driving task under the partially observable Markov decision process (POMDP) setting. POMDP is commonly used to model decision-making problems in nondeterministic and partially observable scenarios.

Similar to (Kurniawati, 2022), the POMDP model is defined as a 5-tuple $< \mathcal{S}, \mathcal{A}, \mathcal{O}, \mathcal{T}, \mathcal{F} >$, where $\mathcal{S}$ is the state space, $\mathcal{A}$ the action space, $\mathcal{O}$ the imitator's observation space, $\mathcal{T}(\boldsymbol{s}_{t+1}|\boldsymbol{s}_t, \boldsymbol{a}_t)$ denotes the state transition function, and $\mathcal{F}(\boldsymbol{o}_{t+1}|\boldsymbol{s}_t, \boldsymbol{a}_t)$ denotes the observation function. Here, variables' subscripts represent the timesteps they are in. We use boldface with lowercase letters to denote instances in corresponding spaces, such as $\boldsymbol{s}_t \sim \mathcal{S}$, $\sim$ denotes the sample process.

The partial observability in the driving task is represented by the mismatch between the expert's and the imitator's observation form. While the expert receives and processes the ground truth information (e.g. bird-eye view images or vectored description of the whole scenario) as state $\boldsymbol{s}$, the

imitator can only observe an image $\boldsymbol{o}$ as the profile of the current expert state, and $\boldsymbol{o}$ only carries partial information about $\boldsymbol{s}$.

Additionally, we consider the driving task under the conditional imitation learning (CIL) setting proposed by (Codevilla et al., 2018), and introduce a measurement vector $\boldsymbol{v}$ that come with $\boldsymbol{o}$. $\boldsymbol{v}$ describes the ego vehicle state and navigation information, and is also provided to the imitator as conditions. As human drivers often drive under the indication of navigation information, we also design our method to express the effect of $\boldsymbol{v}$ more comprehensively to the imitator. In the implementation, the navigation information can be derived from a simple route planner which requires no parameter optimization, and the incorporation of $\boldsymbol{v}$ in CIL does not violate the general end-to-end driving setting.

At each timestep $t$ the expert observes the state $\boldsymbol{s}_t$, selects an action $\boldsymbol{a}_t \sim \pi_e(\boldsymbol{a}_t|\boldsymbol{s}_t)$ based on its policy $\pi_e$, and then observe the next state $\boldsymbol{s}_{t+1} \sim \mathcal{T}(\boldsymbol{s}_{t+1}|\boldsymbol{s}_t, \boldsymbol{a}_t)$ sampled from the state transition function. During the above interaction with environment, image observation $\boldsymbol{o}_t$ and measurements $\boldsymbol{v}_t$ that obtained from the observation function $(\boldsymbol{o}_t, \boldsymbol{v}_t) \sim \mathcal{F}(\boldsymbol{o}_t, \boldsymbol{v}_t|\boldsymbol{s}_{t-1}, \boldsymbol{a}_{t-1})$ are recorded as the proxy of the state $\boldsymbol{s}_t$ observed by expert. The dataset $\mathbf{D_e}$ is organized as a combination of $N$ expert demonstrations $(\boldsymbol{o}_i, \boldsymbol{v}_i, \boldsymbol{a}_i)_{i=1}^N$. The goal of typical BC approaches is to learn a policy $\pi(\boldsymbol{a}_t|\boldsymbol{o}_t, \boldsymbol{v}_t)$ with the supervision of the expert demonstrations such that the distribution of hidden reward generated by the policy $\pi$ is the same as the one generated by expert policy $\pi_e$.

Following the setting of typical Behavioral Cloning from Observation Histories (BCOH) approaches (Chuang et al., 2022; Hu et al., 2022a; Seo et al., 2023), we assume that observations in history can provide useful information about $\boldsymbol{s}_t$, as a snapshot typically cannot tell the whole story. Therefore, we extend the temporal input range of $\pi$ to encourage it to extract more information from the past and have a better understanding of the current scenario. By setting a history perception window length $l$, we use $\boldsymbol{o}_{t-l:t}$ to denote the observed images in the period $[t - l, t]$, $\boldsymbol{v}_{t-l:t}$ is defined in a similar way. In a certain timestep $t$, the observation history under the perception window length $l$ of an imitator is defined as $\boldsymbol{h}_t = (\boldsymbol{o}_{t-l:t}, \boldsymbol{v}_{t-l:t})$, the imitator's policy is rewrote $\pi$ as $\pi(\boldsymbol{a}_t|\boldsymbol{h}_t)$, and the training dataset is organized as $\mathbf{D_e} = (\boldsymbol{h}_i, \boldsymbol{a}_i)_{i=1}^N$.

While the incorporation of observations in the history provides vast information and helps imitators learn the dynamics of the environment, it also introduces shortcuts in imitating and prevents imitators from faithfully recovering $\pi_e$, as showcased by Figure 1a.

## 3 METHOD

### 3.1 INSIGHTS FROM CAUSALITY

We use the causal diagram to give a description of the driving tasks, as shown in Figure 1b. In the modeled causal graph, it's clear to see that the observed variable tuple $(\boldsymbol{o}_t, \boldsymbol{v}_t)$ has not directed edges that point at the current state $\boldsymbol{s}_t$ and action $\boldsymbol{a}_t$. We design this based on the fact that $(\boldsymbol{o}_t, \boldsymbol{v}_t)$ is just a profile of the state observed by the expert, derived as $(\boldsymbol{o}_t, \boldsymbol{v}_t) \sim \mathcal{F}(\boldsymbol{o}_t, \boldsymbol{v}_t|\boldsymbol{s}_{t-1}, \boldsymbol{a}_{t-1})$. Therefore, directly building a policy that maps $(\boldsymbol{o}_t, \boldsymbol{v}_t)$ to $\boldsymbol{a}_t$ is inappropriate, as the state $\boldsymbol{s}_t$ that the expert used for the decision has not been inferred, and $(\boldsymbol{o}_t, \boldsymbol{v}_t)$ cannot cover enough information for replicating $\pi_e(\boldsymbol{a}_t|\boldsymbol{s}_t)$. On the other hand, building an imitator policy $\pi(\boldsymbol{a}_t|\boldsymbol{h}_t)$ that considers historical information can also foster the negative effects of previous actions on identifying the expert's decision process.

In this paper, we propose Causality-inspired Contrastive Conditional Imitation Learning (3CIL), an BC approach that incorporates causal insights into its design. In the training stage, 3CL decomposes the task of IL into two sub-tasks: representation learning and policy learning, corresponding to learning a representation model $G(\hat{\boldsymbol{s}}|\boldsymbol{h})$, and a predictor model $J(\hat{\boldsymbol{a}}|\hat{\boldsymbol{s}})$. Here, we add hats '$\hat{}$' to the imitator's predictions to distinguish them from the expert's states and actions. Based on the causal graph and analysis we made above, we conclude the important traits (**T1,T2**, **T3**) that a robust imitator must have, and introduce the targeted treatments in our proposed method 3CIL, as shown in following paragraphs.

**(T1) Ability to extract enough information from observation history, bridging a mapping** $\boldsymbol{h}_t \rightarrow \hat{\boldsymbol{s}}_t$ **matches** $(\boldsymbol{s}_{t-1}, \boldsymbol{a}_{t-1}) \rightarrow \boldsymbol{s}_t$: a robust imitator must have its clues about the current scenario. To achieve **T1**, 3CIL imposes a future image reconstruction task on its representation learning

phase. With emphasis on the temporal and navigation conditions, we propose to model the feature extraction module with a conditional Variational Auto-Encoder (VAE) and recurrent process. Based on supervision from the causal direction $\hat{s}_t \rightarrow \hat{o}_{t+1}$, 3CIL transforms the history concluding process into a simulation of observation function $\mathcal{F}(o_{t+1}|s_t, a_t)$ in the modeled POMDP. With this modification, the representation model $G(\hat{s}|h)$ is urged to conclude enough information from observation history $h_t$, so that it can match with the actual expert state $s_t$ in the metric of quality of inferred future image observation $\hat{o}_t$.

**(T2) Minor reliance on spurious correlations, learning influence of previous actions through $a_{t-1} \rightarrow s_t \rightarrow a_t$ instead of $h_t \leftarrow a_{t-1} \rightarrow a_t$:** to guarantee its performance in non-i.i.d deployment time. To accomplish **T2**, 3CIL incorporates an action residual prediction task in the representation learning phase to encourage the model to capture changes in the expert's decisions. The proposed action residual prediction task enforces the imitator to capture the variations within consecutive actions $\Delta a_t = a_t - a_{t-1}$ without explicitly accessing $a_t$ and $a_{t-1}$, which bypasses spurious correlation by the effect estimation in causal direction $\hat{s}_t \rightarrow \Delta \hat{a}_t$ (i.e., require $s_t$ to reflect clues about changes between actions, instead of serve as proxy of $a_{t-1}$). We also add a contrastive learning objective to help shape a regression-aware representation space that provides clues about current action. The supervised contrastive learning guides the representation model with the anti-causal direction $\hat{s}_t \leftarrow a_t$ hindsight, to enhance the consistency of causal effect from $\hat{s}_t$ to predicting action $a_t$.

**(T3) Ability to investigate the difference between the expert's policy and the imitator's policy, identifying scenarios that caused high divergence between $a_t$ and $\pi(a_t|h_t)$:** a robust imitator must not be satisfied with its great average offline evaluation performance, but to focus on the factors that caused its inconsistency with expert behaviors. For **T3**, 3CIL proposes a sample weight term to guide the imitator focusing on samples that cause their predictions to contradict the expert.

In conclusion, the above three traits indicate stable causal influences: $h_t \rightarrow \hat{s}_t$ and $\hat{s}_t \rightarrow \hat{a}_t$, to simulate the actual expert strategy. Figure 2 and Section 3.2 describe the treatments used in 3CIL that enhance the robustness of the representation model $G(\hat{s}|h)$ from both causal direction ($\hat{s}_t \rightarrow \Delta \hat{a}_t, \hat{s}_t \rightarrow \hat{o}_{t+1}$) and anti-causal direction ($\hat{s}_t \leftarrow a_t$). With supervision from these two directions, the extracted representation $\hat{s}_t$ is impulsed to have steady causal effects on the descendant nodes of actual expert state $s_t$, help aligning inferred state $\hat{s}_t$ with $s_t$ to produce a stable representation. Figure 3 and Section 3.3 illustrate the optimizing process of predictor model $J(\hat{a}|\hat{s})$. The incorporated sample-weighting term is inspired by classical studies in causal reasoning, to mitigate the biases in both the representation learning stage and expert demonstration distribution. We detail the design of learning objectives in 3CIL in the following sections.

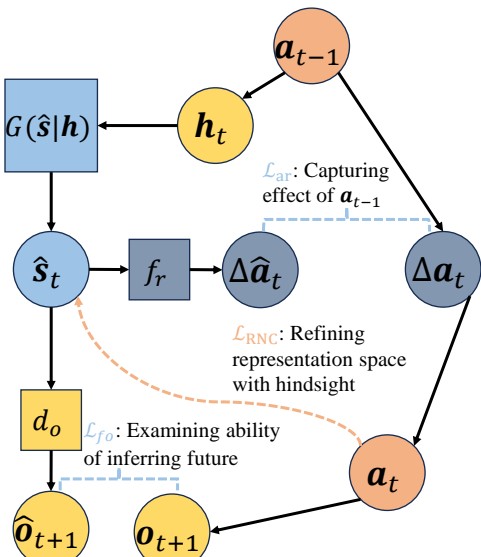

Figure 2: Modules in our 3CIL method are represented by rectangles, and colored based on the type of variables they are about to predict. 3CIL optimizes its representation model $G(\hat{s}|h)$ from both causal direction ($\mathcal{L}_{\text{ar}} : \hat{s}_t \rightarrow \Delta \hat{a}_t, \mathcal{L}_{fo} : \hat{s}_t \rightarrow \hat{o}_{t+1}$) and anti-causal direction $\mathcal{L}_{\text{RNC}} : \hat{s}_t \leftarrow a_t$. Here, $f_r$ is the action residual predictor: $\Delta \hat{a}_t = f_r(\hat{s}_t)$, and $d_o$ is an image decoder predicts the image in next frame : $\hat{o}_{t+1} = d_o(\hat{s}_t)$ .

## 3.2 REPRESENTATION LEARNING

The idea of representation learning is to train a representation model $G(\hat{s}|h)$ that extracts meaningful and reliable features for downstream predictor, corresponding to accomplish both **T1** and **T2**. Appendix A.2.1 provides an introduction to the implementation details and a visualization of modules in $G(\hat{s}|h)$.

As shown in (Codevilla et al., 2018), CIL eases the driving task by introducing conditions, i.e. navigation information organized as route commands that indicate what high-level action the agent

should take in the current route (e.g. lane following, turn left, turn right), into the input field of policy. Here, we move a step forward: instead of appending route commands into the feature vector for downstream predictor, 3CIL processes the route command as factors that affect the state transition.

Inspired by approaches in causal inference (Nie et al., 2021; Schwab et al., 2020) that impose the influence of treatment variables into networks' parameters for more accurate estimations, we seek to integrate the effect of navigation condition in the whole process of the representation model. By embedding the measurement vector $v$ (including ego vehicle speed, and route commands) into a feed-forward network that decides the parameters of the hidden state's posterior distribution, we merge the influence of navigation information and current speed into the state abstraction process. Therefore, by using these navigation conditions as proxy variables of actual states observed by only experts, 3CIL guides the representation model to have a more reliable estimation of the current state.

Concretely, we design our $G(\hat{s}|h)$ as a conditional VAE, with a recurrent state sequence module (RSSM) proposed by (Hafner et al., 2019), to simulate the state transition function $\mathcal{T}(s_{t+1}|s_t, a_t)$. We first extract the dense features $(x_{t-l:t}, m_{t-l:t})$ from non-structured historical images $o_{t-l:t}$ and raw measurement vectors $v_{t-l:t}$, with a pre-trained image encoder $E_o$ and a measurement vector encoder $E_v$ in $G(\hat{s}|h)$, produced as $(x_{t-l:t}, m_{t-l:t}) = [E_o(x_{t-l:t}|o_{t-l:t}), E_v(m_{t-l:t}|v_{t-l:t})]$. After that, we model the latent representation $\hat{s}_t$ as conditioned on these features.

As empirically shown in (Hu et al., 2022a; Hafner et al., 2019), using both deterministic and stochastic features to model the latent representation enhances the flexibility and capability of the representation model. Therefore, we design the estimated latent state in timestep $t$: $\hat{s}_t$ as a combination of the deterministic historical features $c_t$ and the stochastic current latent information $z_t$. RSSM models the distribution of current latent information $z_t$ under transition $\mathcal{T}$ by a posterior distribution $q_z(z_t|c_t, m_t, x_t) : z_t \sim \mathcal{N}(\mu_\theta(c_t, m_t, x_t), \sigma_\theta(c_t, m_t, x_t))$ that conditioned on historical information $c_t$, and features $(m_t, x_t)$ that derived from measurement vector and image observation. Here, the mean $\mu_\theta$ and standard deviation $\sigma_\theta$ of modeled Gaussian distribution $q_z$ are predicted by a feed-forward network that takes $(c_t, m_t, x_t)$ as input. The historical features $c_t$ are extracted from a recurrent network $f_d$ that takes former historical information $c_{t-1}$ and former latent information $z_{t-1}$ as input. For a training set $\mathbf{D_e}$ with $N$ samples, the optimization objective is written in a variational lower bound form:

$$ELBO = \sum_{i=1}^{N-1} \underbrace{\mathbb{E}_{G(z_i, c_i|h_i)}[log p(o_{i+1}|z_i, c_i)]}_{\mathcal{L}_{fo}: \text{ future image reconstruction}} - \underbrace{D(q_z(z_i|c_i, m_i, x_i)||p_z(z_i|c_{i-1}, z_{i-1}))}_{\text{posterior regularization}}, \quad (1)$$

where $D(\cdot||\cdot)$ denotes the KL-divergence measurement, and the future image reconstruction task $\mathcal{L}_{fo}$ is carried out by an image decoder that receives $(z, c)$ as input. Unlike previous works that reconstruct all images in $h_t$ to examine the ability of $G(\hat{s}|h)$ in reserving all history information, we resort to future image reconstruction $\mathcal{L}_{fo}$ to evaluate abilities of $G(\hat{s}|h)$ to extract history information and infer future state. With this modification, $\mathcal{L}_{fo}$ indicates the representation model $G(\hat{s}|h)$ to capture temporal evolution by examining the simulated future image $\hat{o}_{t+1}$. Therefore, $\hat{s}$ is urged to reproduce the similar causal effect of $s_t$ on $o_{t+1}$.

Different from previous approaches (Hu et al., 2022a; Hafner et al., 2019), we eliminate the explicit use of the previous action variable $a_t$ in all modules in the representation model, which seems to contradict the causal effect $a_{t-1} \rightarrow s_t$ modeled in the causal diagram Figure 1b. However, the causal diagram further shows that the effect of previous action $a_{t-1}$ on current action $a_t$ can be inferred from the variation between them, denotes as $\Delta a_t = a_t - a_{t-1}$. In this view, we proposed to maximize the conditional mutual information $I(\hat{s}_t, a_t|a_{t-1})$, by optimizing the prediction accuracy of $\Delta a_t$ by $\hat{s}$ only. Therefore, we introduce an action residual prediction objective in the representation learning phase:

$$\mathcal{L}_{ar} = \frac{1}{N} \sum_{i=1}^{N} (\Delta a_i - f_r(\hat{s}_i))^2, \quad (2)$$

where $f_r$ is an action residual predictor that predicts action difference $\Delta a_t = a_t - a_{t-1}$. The introduced action residual prediction task builds a causal directed edge $\hat{s}_t \rightarrow \Delta \hat{a}_t$ to encourage $G(\hat{s}|h)$ capturing variations caused by previous actions.

While embedding the influence of observation history and action residual into the representation learning enhances the model's ability to infer the current state, it inevitably introduces more prominent spurious correlations, as discussed in former sections. Therefore, to alleviate such effects, we further introduce a contrastive learning objective to help shape a robust representation model.

As we aim to achieve driving with expert demonstrations, we resort to the supervised contrastive learning (Khosla et al., 2020) methods to enhance robustness, as the expert's actions naturally reflect differences among samples. With this intuition, we introduce the Rank-N-Contrast (RNC) loss from (Zha et al., 2023) to the optimization objective of representation learning in 3CIL. The idea of RNC is to align distances in the representation space ordered by distances in their labels, which by design meets the requirement of alleviating the effect of spurious correlation left in the features: samples that carry similar historical information may end up poles apart in the representation space when their corresponding action labels are different.

Therefore, $G(\hat{s}|h)$ equipped with the RNC loss is encouraged to use the anti-causal relation $\hat{s}_t \leftarrow a_t$ to infer a suitable state corresponding to the actual expert action $a_t$. In this view, the performed supervised contrastive learning is equivalent to conducting a hindsight investigation on the consistency of constructed causal effect $\hat{s}_t \rightarrow a_t$.

For a batch sampled from $\mathbf{D_e}$ with batch size $B$, we apply two independent augmentations to obtain a new batch with $2B$ samples, and write the RNC loss as:

$$\mathcal{L}_{\text{RNC}} = \frac{1}{2B} \sum_{i=1}^{2B} \frac{1}{2B-1} \sum_{j=1, j \neq i}^{2B} -\log \frac{\exp(\text{sim}(\hat{s}_i, \hat{s}_j)/\tau)}{\sum_{\hat{s}_k \in set(i,j)} \exp(\text{sim}(\hat{s}_i, \hat{s}_k)/\tau)} , \tag{3}$$

where $\tau$ is the temperature parameter, $\text{sim}(\cdot)$ is the similarity measure between two inferred states (cosine similarity is used in this work), $set(i,j)$ collect those samples' representations whose corresponding action labels have higher rank (in terms of distance with $\hat{s}_i$) compare to $\hat{s}_j$ (i.e., $set(i,j) = \hat{s}_k | k \neq i, \text{d}(a_k, a_i) \geq \text{d}(a_j, a_i), \text{d}(\cdot, \cdot)$ measures distance between two labels).

After introducing all designs we proposed that aim to shape a robust representation model which meets the requirements **T1** and **T2**, we write our final optimization objective of $G(\hat{s}|h)$ as maximizing: $\mathcal{L}_G = ELBO - \mathcal{L}_{\text{ar}} - \mathcal{L}_{\text{RNC}}$.

### 3.3 POLICY LEARNING

After representation model $G(\hat{s}|h)$ is trained and its parameters are frozen, the downstream predictor network $J(\hat{a}|\hat{s})$ is more resistant to spurious correlations. However, a severe problem remains unsolved: the inconsistency of imitators when replicating an expert's strategy in certain scenarios. We believe this problem can be attributed to the unmatched growth between the diversity of driving scenarios and the number of expert demonstrations: scenarios are not distributed uniformly in the dataset. Such a characteristic indulges the imitator to be indifferent about rare scenarios and hinders it from achieving **T3**.

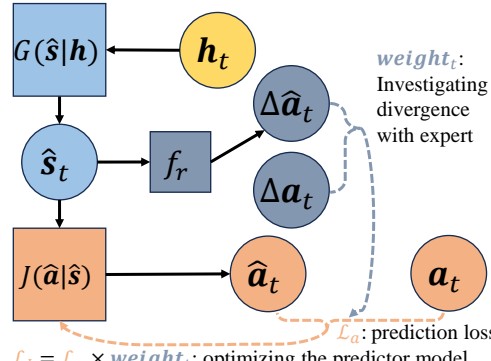

Figure 3: Modules in our 3CIL method are represented by rectangles, and colored based on the type of variables they are about to predict. 3CIL incorporates the sample-weighting term $\boldsymbol{weight}_t$ into the optimization objective of predictor model $J(\hat{a}|\hat{s})$, to transform the divergence between the expert and imitator into emphasis on important samples.

To enhance the imitator's ability to cope with rare situations, 3CIL incorporates a sample-weighting process in training the predictor $J(\hat{a}|\hat{s})$: i.e., we assign weights on samples based on the divergence between imitator and expert, enforcing the predictor attaching importance on rare situations. To detect the extent of divergence, we reuse the action residual predictor $f_r$ that was trained from the previous phase, to describe such difference in a per-sample manner.

Denote the action residual prediction error between $f_r(\hat{s})$ and $\Delta a$ as $\delta a$ : $\delta a = |\Delta a - f_r(\hat{s})|$, and denote the operation: $\mathrm{bound}(\cdot, b_{min}, b_{max})$ as a function bounds a variable or all elements within a vector, to the range $[b_{min}, b_{max}]$. For a sample $(h_i, a_i)$ from expert demonstration, its corresponding weight is computed as:

$$\boldsymbol{weight}_i = \exp(\mathrm{bound}(\delta a_t - \overline{\delta a}, b_{min}, b_{max}) \times \gamma), \tag{4}$$

where $\overline{\delta a}$ is the mean of residual errors in the minibatch, $[b_{min}, b_{max}]$ is set to $[-0.3, 0.3]$, $\gamma$ is the factor controlling strength of sample weight and it is set to $6.67$ in our experiment.

Consider the causal relations between $(s_t, \Delta a_t, a_t)$ in Figure 1b, variations in $\delta a_t$ can be seen as performing interventions on $\Delta a_t$, such variation based sample-weighting process is akin to techniques like inverse probability weighting and doubly robust learning that widely used in the causal inference literature. Indeed, the quantification of prediction error for $\Delta a_t$ identifies the potentially under-represented certain scenarios and also mediates the bias introduced by inaccurate estimations in previous representation learning.

The training objective of predictor $J(\hat{a}|\hat{s})$ is then to minimize: $\mathcal{L}_J = \frac{1}{N}\sum_{i=1}^N \mathcal{L}_a(a_t, J(\hat{s}_i)) \times \boldsymbol{weight}_i$, where $\mathcal{L}_a$ can be typical metrics that used for supervised learning (e.g., cross-entropy and mean-squared error, based on the form of expert action label).

3CIL divides the imitating process as two separate stages: first use $\mathcal{L}_G$ to train a representation model $G(\hat{s}|h)$, then $\mathcal{L}_J$ is used to train a predictor model $J(\hat{a}|\hat{s})$ while the parameters of $G(\hat{s}|h)$ is frozen. We detail the implementation of 3CIL method in Appendix A.2. For a certain sample $h_i$, the imitator's prediction is made as: $\hat{a}_i \sim J(\hat{a}_i|\hat{s}_i), \hat{s}_i \sim G(\hat{s}_i|h_i)$.

## 4 EXPERIMENTAL EVALUATION

### 4.1 SETTINGS

**Environments and dataset:** We conduct in the visually complex driving simulator CARLA (Dosovitskiy et al., 2017). Expert demonstrations are collected by an RL agent from (Zhang et al., 2021b) that trained using privileged information as input, we deploy it in four towns (Town01, Town03, Town04, Town06) to generate about $N = 1,125,300$ training samples. These samples is then organized in form of $\mathbf{D_e} : (h_i, a_i)_{i=1}^N$, with the perception window length $l$ set to $4$, and the observation history $h_t$ is organized as $h_t = (o_{t-l:t}, v_{t-l:t})$. An image observation $o$ is of size (channel=RGB, width=240px, height=150px), and a measurement vector $v$ is composed as $v = (v_{speed}, v_{route\_command}, v_{route\_command\_next})$.

In the testing phase, we modify the weather conditions, traffic density, camera parameters, and also introduce two new towns (Town02, Town05) as environments to evaluate the performance of methods under severe distribution shifts. We design four evaluation scenario settings (denote as Scenario 1,2,3,4) corresponding to experiments in four towns that are seen in $\mathbf{D_e}$, and two evaluation scenario settings (denote as Scenario 5,6) for experiments in Town02 and Town05. Appendix A.3.3 and A.3.4 described the experiment design in detail.

**Baselines:** For comparison, we choose the following baselines as representatives of different types of approaches. (1) Conditional Imitation Learning (**CIL**, (Codevilla et al., 2018) ): stands as the vanilla CIL method. (2) Keyframe-Focused Visual Imitation Learning (**Keyframe**, (Wen et al., 2021)): utilizes action prediction errors to assign weights on samples. (3) Domain Generalizable Imitation Learning by Causal Discovery (**DIGIC**, (Chen et al., 2024)): performs causal discovery to sort causal features and learn a domain generalizable policy. (4) Past Action Leakage Regularization (**PALR**, (Seo et al., 2023)): imposes regularization on conditional dependence between inferred state $\hat{s}_t$ and previous action $a_{t-1}$ to alleviate spurious correlation. (5) **Premier-TACO** from (Zheng et al., 2024): conducts temporal action-driven contrastive learning to shape a robust representation model. We also implement two methods that can be seen as ablation experiments of 3CIL. (6) Rank-N-Contrast framework (**RNC**, (Zha et al., 2023)): adding the $\mathcal{L}_{RNC}$ loss into the training representation model, without action residual prediction task. (7) Visual Imitation Learning via Residual Action Prediction (**RAP**, (Chuang et al., 2022)): adding the $\mathcal{L}_{ar}$ in training representation, without assigning

sample weights in training predictor.A detailed introduction about used baselines and the ways we implement them is provided in Appendix A.3.1.

**Evaluation metrics:** We quantify the performance of methods based on three metrics: accumulated rewards, average collision rate, and average speed. The reward function is constructed as $R = r_{speed} + r_{position} + r_{rotation} + r_{action}$, a combination of four factors that independently judge an agent's driving ability in following indicated routes, similar to previous work (Zhang et al., 2021b). Details of the reward function are listed in Appendix A.3.5. We combine all metrics to discuss strategies learned by each method.

## 4.2 PERFORMANCE AND DISCUSSION

Table 1: Performance of each method in three metrics: accumulated reward (R), average collision rate (C, in ‰), and average speed (S, in km/h). We add arrows beside R and C to indicate the optimal direction. No arrow is placed beside S, as the speed metric alone can not reflect driving performance. **Bold** numbers in rows R and C indicate the best results, second-best results are underlined.

| Method / Metric | | CIL | Keyframe | DIGIC | PALR | Premier-TACO | RNC | RAP | 3CIL(Ours) |
|---|---|---|---|---|---|---|---|---|---|
| Scenario 1 | R ↑ | 330.49 | 353.83 | 437.32 | 354.47 | 469.99 | 411.31 | 383.54 | **521.26** |
| | C ↓ | 0.66 | 0.58 | 0.73 | 2.42 | 0.85 | 0.55 | 0.60 | **0.54** |
| | S | 5.22 | 6.05 | 11.99 | 8.52 | 12.59 | 7.50 | 7.95 | 9.76 |
| Scenario 2 | R ↑ | 12.14 | 309.70 | 484.49 | 422.79 | 431.22 | 519.14 | 362.31 | **587.44** |
| | C ↓ | **0.36** | 0.57 | 0.47 | 1.67 | 0.55 | 0.42 | 0.53 | 0.46 |
| | S | 7.89 | 9.56 | 18.96 | 19.11 | 19.76 | 18.12 | 15.95 | 19.85 |
| Scenario 3 | R ↑ | 247.29 | 125.30 | 404.44 | 327.85 | 204.38 | 64.80 | 136.6 | **420.38** |
| | C ↓ | 1.35 | 1.56 | 1.38 | 4.18 | 1.31 | 2.20 | 3.15 | **1.25** |
| | S | 3.99 | 7.74 | 13.43 | 9.60 | 10.76 | 12.67 | 11.66 | 12.07 |
| Scenario 4 | R ↑ | 345.00 | 529.68 | 400.42 | 837.98 | 561.13 | 735.19 | 505.63 | **966.35** |
| | C ↓ | 0.37 | 0.31 | 0.32 | 0.97 | 0.37 | 0.31 | 0.35 | **0.27** |
| | S | 7.00 | 9.46 | 18.58 | 11.23 | 19.40 | 17.05 | 16.78 | 16.20 |
| Scenario 5 | R ↑ | 7.18 | 278.95 | 306.10 | 421.16 | 516.70 | 299.72 | 302.50 | **538.50** |
| | C ↓ | **0.29** | 0.53 | 0.49 | 1.47 | 0.37 | 0.50 | 0.59 | 0.48 |
| | S | 7.61 | 9.24 | 12.60 | 11.92 | 15.32 | 13.71 | 11.00 | 14.46 |
| Scenario 6 | R ↑ | 45.93 | 215.77 | 409.88 | 389.07 | 331.29 | **447.44** | 195.53 | 447.24 |
| | C ↓ | **0.34** | 0.64 | 0.94 | 1.54 | 0.68 | 0.64 | 0.63 | 0.59 |
| | S | 4.32 | 8.95 | 10.98 | 8.66 | 12.19 | 8.56 | 7.78 | 10.99 |

In Table 1, we present the evaluation results in the CARLA simulator. A detailed ablation study is provided in Appendix A.4. We analyze the performance of methods from several aspects.

**Effect of spurious correlation.** Our proposed method 3CIL is one of the most cautious drivers with the lowest collision rate in half settings (3 of 6). Interestingly, another method with the lowest collision rate is the earliest approach CIL Codevilla et al. (2018) which did not consider the spurious correlations problem. This phenomenon can be explained when we consider the accumulated reward and average speed: the quantized results of CIL in these two metrics are significantly lower than most of its competitors, suggesting that CIL has built doubtful decision patterns that showed overly cautiousness. Indeed, in our observation, the agent trained with CIL often got stuck or even failed to launch. In contrast, the rest of the methods generally have no such issues, demonstrating the necessity of alleviating the effects of spurious correlations.

On the contrary, PALR effectively removes the effect from the previous action by regularizing conditional dependence $(\hat{s}_t; a_{t-1}|a_t)$. However, such regularization may not be suitable for driving tasks, as shown in Figure 1b, investigation of effects from previous action is still required for recovering information about the expert's state. Indeed, although PALR achieves relatively stable rewards, it obtains the highest collision rates in most settings.

**Sample-weighting strategies and contrastive learning help imitating.** Both Keyframe and 3CIL translate errors in prediction into assigned weights on corresponding samples. Experimental results demonstrate that sample-weighting strategies indicate the imitator to focus on crucial changepoints

in a simple yet efficient way. Moreover, the proposed weighting design in 3CIL utilizes errors in action residual prediction in the representation learning stage instead of the copycat policy action prediction errors in (Wen et al., 2021). Such a strategy enables 3CIL to identify abnormal scenes that the representation model fails to cover and empowers it to cope with more general problems than the copycat phenomenon.

On the other hand, contrastive learning also contributes the imitating performance. Both Premier-TACO (temporal action-driven contrastive loss) (Zheng et al., 2024) and RNC (supervised contrastive loss) (Zha et al., 2023) can assist the representation model to capture the intrinsic characteristics from observation history, which is demonstrated by their relatively good performance in both seen and unseen scenarios. Our 3CIL approach utilizes supervised contrastive learning to help infer states, as the actions made by experts can provide clear clues for constructing representation space, without the need to tune hyper-parameters for sampling temporal positive/negative pairs.

While methods like DIGIC that introduce causal discovery can also investigate the essential relationships related to experts' decisions, the masking or filtering operation will inevitably cause information loss, which may lead to safety concerns in applications like driving, and sub-optimal performance in evaluation.

**Robustness requires effort.** Although variations in experimental settings and novel map layouts pose challenges to imitators' capabilities, 3CIL still maintains a robust driving strategy. As shown in Table 1, 3CIL obtains the highest accumulated rewards in most settings (5 of 6). Recall that the reward function in evaluation measures the ability of an imitator in executing general driving tasks given navigation conditions, the outstanding performance in accumulated rewards indicates that the pursuit for **T1,T2,T3** does improve the robustness of the imitator.

Moreover, when we analyze the performance of RNC and RAP that can be seen as parts of the ablation study of 3CIL, the effectiveness of interactions we introduced in Figure 2 and Figure 3 become evident: the shared reconstruction task $\mathcal{L}_{fo}$, coupled with hindsight from supervised contrastive learning task $\mathcal{L}_{\mathrm{RNC}}$ or action variation capturing task $\mathcal{L}_{\mathrm{ar}}$, both alleviate the spurious correlations, but fail to maintain steady performance in all settings. Concretely, when we incorporate the anti-causal direction $\hat{s} \leftarrow \boldsymbol{a}_t$ influence to shape the imitator's representation space, $G(\hat{s}|\boldsymbol{h})$ is enforced arrange inferred states to match their corresponding potential actions' propensity, but not enforced to capture the effect that previous actions imposed on the current state, while RAP approach is equivalent to do the opposite. Either way, the absence of essential information will result in an inadequate estimation of the expert state, hindering models from robust generalization performance. Also, the sample-weighting process proposed in 3CIL does improve the imitator's ability to handle rare scenarios, as the agent trained with 3CIL shows more caution and obtains lower collision rates.

## 5    CONCLUSION

In this work, we investigate the factors hindering imitation learning methods from generalizing training performance into unfamiliar testing environments in autonomous driving tasks. Based on causal reasoning about the expert's decision process, we identify crucial traits an imitator must have for robust performance. After that, we introduce Causality-inspired Contrastive Conditional Imitation Learning (3CIL), an imitation learning method that imposes regularization on the imitator's representation by supervised contrastive learning and action residual prediction, corresponding to assigning supervisions on representation model from both causal direction and anti-causal direction to guarantee quality of the inferred state. Moreover, 3CIL introduces a sample-weighting term to transform the high divergences between the expert and imitator, into the emphasis on rare scenarios, enabling the imitator to adapt to diverse situations. We perform experiments in the CARLA simulator to demonstrate the effectiveness of the proposed 3CIL.

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

# A APPENDIX

## A.1 RELATED WORK

### A.1.1 IMITATION LEARNING FOR AUTONOMOUS DRIVING

Imitation Learning (IL) is widely used in autonomous driving (Le Mero et al., 2022; Bansal et al., 2018), as it requires few or zero actual interactions with the target environment. Classical literatures have divided IL into behavior cloning (BC) and inverse reinforcement learning (IRL). The idea of adversarial learning also introduces adversarial imitation learning (AIL) to the IL family. End-to-end autonomous driving approaches typically use BC in training, as BC does not need actual interactions with the environment, but seeks to learn driving patterns from numerous offline demonstrations(Chib & Singh, 2023).

However, the performance of BC is often problematic, which arises from the contradiction between the i.i.d assumption made by BC and the out-of-distribution (OOD) nature of driving tasks (Sridhar et al., 2023). Such a conflict leads to the compounding error that a BC imitator behaves unreliably when observing unfamiliar scenarios. Moreover, BC often suffers from causal confusion (de Haan et al., 2019), as its lack of the explicit causal model makes the imitator cannot tell the difference between spurious correlations and causal relations. This phenomenon becomes more severe when the BC imitator interacts with the environment in a sequential manner. Commonly the learned shortcuts (Wen et al., 2022) fail to apply in the test stage, or the BC imitator is even stuck in delusions caused by itself(Ortega et al., 2021).

### A.1.2 REMEDIES FOR CAUSAL CONFUSION

Previous approaches have proposed several remedies for handling the phenomenon of causal confusion to obtain a robust imitator, including: randomly masking encoded discrete features (Park et al., 2021), incorporating additional supervisions or regularization on the encoder or predictor (Hu et al., 2022a; Kumar et al., 2023; Seo et al., 2023), querying experts about certain scenarios (de Haan et al., 2019) and performing interventions on the environment state or policy input (Pfrommer et al., 2023; Ruan & Di, 2022), filtering input features based on causal discovery (Chen et al., 2024; Samsami et al., 2021), maximizing certain bounds or mutual information to achieve deconfounding (Swamy et al., 2022a; Wan et al., 2023).

However, masking and regularizing encourage the imitator to be indifferent about spurious correlated features but may cause the loss of useful information. Adding extra supervision during training requires modifying the data-collecting process, and querying experts or intervening in the environment seems unfeasible in the driving task. While approaches with causal discovery match humans' instinct, they typically only work with low-dimensional data form (i.e., vectored observation), and the choice of causal discovery algorithm impacts the imitator's performance. Approaches aimed to deconfound may need more clues about the expert's policy and the dynamic of target environments or mainly contribute to the theoretical analysis. Therefore, we aim to develop a method that requires no extra information beyond the training dataset and has a robust policy that can drive in unseen environments without reliance on spurious correlations.

### A.1.3 CAUSAL REASONING

Causal reasoning (Pearl, 2009) approaches can be generally divided into two genres: causal discovery and causal inference. Causal discovery aims to recover the underlying causal relations among variables in the target system, to help researchers learn the mechanisms of a system and aid downstream tasks. On the other side, causal inference is designed to learn the effect of modifying one/multiple variables' value (i.e., intervening on treatment variables) on the outcome variables, while considering the mechanisms between variables.

While typical studies have demonstrated the effectiveness and necessity of causality, incorporating causal reasoning into visually complex and partially observable tasks is still challenging. While efforts have been made to investigate causality in high-dimensional and confounded data (Günther et al., 2023; Zhu et al., 2022; Wang & Zhou, 2021; Wang et al., 2021; Sun et al., 2021; Cheng et al., 2022; Yang et al., 2021), the full identification of causalities in tasks like autonomous driving is still intractable without further assumptions (Zheng et al., 2018) or specifications (Huang et al., 2020). Moreover, using the observational samples alone generally cannot provide sufficient indications for recovering all causal relations or estimating precise causal influence. Performing interventions (Pfrommer et al., 2023; Zhang et al., 2021a) or querying experts about certain scenarios (de Haan et al., 2019; Zhang & Cho, 2017) are unfeasible in tasks with safety concerns and high interaction frequency like robotics and autonomous driving.

### A.2 IMPLEMENTATION OF 3CIL

### A.2.1 MODULES

As shown in Figure 4, $G(\hat{s}|h)$ is composed of an image encoder $E_o(x|o)$, a measurement vector encoder $E_v(m|v)$, a recurrent state sequence module (RSSM) from (Hafner et al., 2019) that combines both deterministic state model $f_d(c_t|c_{t-1}, z_{t-1})$ and stochastic state model $q_z(z_t|c_t, x_t, m_t)$. $x_t$ and $m_t$ are the features extracted from the image encoder $E_o$ and the measurement vector encoder $E_v$, respectively. $c_t$ is a feature vector that preserves history information, and $z_t$ is the hidden state sampled from a Gaussian distribution $p_s(z_t|c_t, x_t, m_t)$ whose mean and variance are parameterized by a feed-forward network.

Denote $\oplus$ as the concatenation operation, for a sample in timestep $t$ with observation history $h_t$, its corresponding representation is obtained as: $\hat{s}_t = c_t \oplus z_t$, $c_t \sim f_d(c_t|c_{t-1}, z_{t-1})$, $z_t \sim q_z(z_t|c_t, x_t, m_t)$, and we assume a fixed initial feature vector $c_0$ without loss of generality.

In addition, we incorporate an image decoder $d_o(\hat{o}_{t+1}|\hat{s}_t)$ to carry out the image reconstruction task, and an action residual predictor $f_r(\hat{\Delta}a_t|\hat{s}_t)$ to capture the variation in expert's actions in a period. In

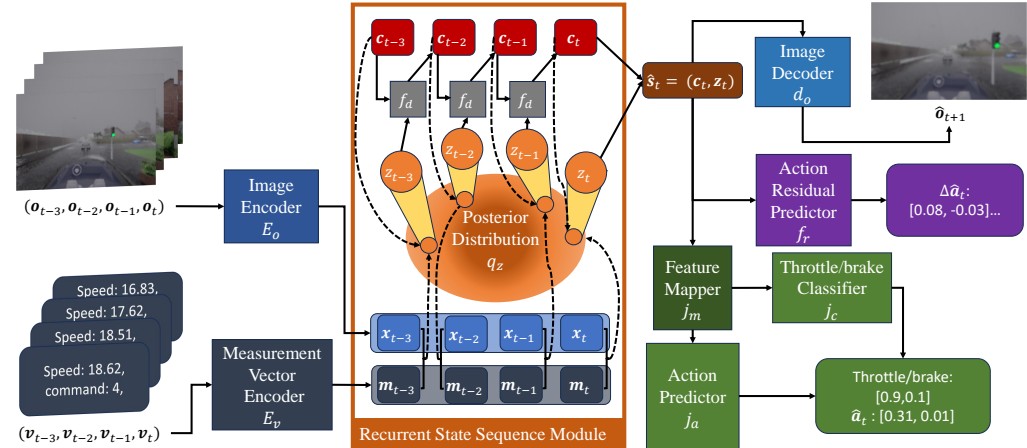

Figure 4: The illustration of proposed framework in 3CIL, with observation history perception window length $l$ set to 3 as example. Dashed edges denote the sampling process. The representation model $G(\hat{s}|\boldsymbol{h})$ is composed of an image encoder $E_o$, a measurement vector encoder $E_v$, and a recurrent state sequence module $(f_d, q_z)$. The predictor model $J(\hat{\boldsymbol{a}}|\hat{s})$ is composed of a feature mapper $j_m$, a throttle-or-brake classifier $j_c$ and an action predictor $j_a$.

practice, we use the Mean-Squared Error (MSE) loss to carry out the maximum likelihood estimation for future image reconstruction part in Eq.1. MSE loss is also used in optimizing the action residual prediction accuracy, as shown in Eq.2.

Similar to approaches (Cultrera et al., 2023) that divide the action command prediction task into multi-stages to restrain the inertia problem, our predictor model $J(\hat{\boldsymbol{a}}|\hat{s})$ is composed of a throttle-or-brake classifier $j_c$ and an action predictor $j_a$.

The classifier $j_c(P(go), 1 - P(go)|\hat{s})$ processes the coarse action command corresponding to go and stop as a binary classification task, $P(go)$ denotes the probability of increasing speed, while $1 - P(go)$ represents the probability of slowing down.

The prediction of $j_c$ is then fed into $j_a$ as part of the action predictor's input features. $j_a(\hat{\boldsymbol{a}}|P(go), 1-P(go), \hat{s})$ predicts the steering angle and the absolute value of acceleration $[\hat{a}_{steer}, \text{abs}(\hat{a}_{acc})]$, the final prediction of $J(\hat{\boldsymbol{a}}|\hat{s})$ is then: if $P(go) > 0.5$, $\hat{\boldsymbol{a}} = [\hat{a}_{steer}, \text{abs}(\hat{a}_{acc})]$, else $\hat{\boldsymbol{a}} = [\hat{a}_{steer}, -\text{abs}(\hat{a}_{acc})]$. We use binary cross-entropy loss to optimize $j_c$, and MSE loss to optimize $j_a$. These two loss terms are then multiplied with the sample-weighting term Eq. 4 to produce the final action loss.

### A.2.2 SPECS

We list the implementation details of modules in 3CIL in this section. Table 2 concludes the structures of data and major components of 3CIL.

Starting with the representation model $G(\hat{s}|\boldsymbol{h})$, the image encoder $E_o(\boldsymbol{x}|\boldsymbol{o})$ is implemented as a pre-trained ResNet18 model (He et al., 2016), while the measurement vector encoder $E_v(\boldsymbol{m}|\boldsymbol{v})$ is a multi-layer perceptron (MLP) coupled with embedding layers that process discrete navigation commands. $v_{route\_command}$ and $v_{route\_command\_nex}$ are processed through an embedding layer (embedding_num = 7, embedding_dim = 8), then concatenate with $v_{speed}$ and feed to a MLP (linear layers = 3, hidden_units = 128, output_dim = 128, activiation = ReLU($\cdot$)) to produce the encoded feature $\boldsymbol{m}$.

In implementation, the RSSM model of $G(\hat{s}|\boldsymbol{h})$ is composed of a linear layer that maps the feature $\boldsymbol{x}_{t-l:t} \oplus \boldsymbol{m}_{t-l:t}$ extracted from $E_o$ and $E_v$, into a vector with size = 256, a GRU module whose both input size and hidden size set to 256 is added as the instance of recurrent network $f_d$, a MLP (linear layers = 3, hidden_units = 256 + 128, output_dim = 128 $\times$ 2, activiation = ReLU($\cdot$)) is used to predict the mean and standard deviation of the posterior distribution $q_z(\boldsymbol{z}_t|\boldsymbol{c}_t, \boldsymbol{m}_t, \boldsymbol{x}_t)$. Finally, a

Table 2: Structures of data and models.

| Type | | Specification |
|---|---|---|
| Observation | | $\boldsymbol{o}_t$ = (channel=RGB, width= 240px, height=150px) |
| Condition | | $\boldsymbol{v}_t = (v_{speed} \sim [0, 100],$ $v_{route\_command} \in \{-1, ..., 6\},$ $v_{route\_command\_next} \in \{-1, ..., 6\})$ |
| Input History | | $\boldsymbol{h}_t = (\boldsymbol{o}_{t-l:t}, \boldsymbol{v}_{t-l:t}), l = 4$ |
| Representation Model $G(\hat{\boldsymbol{s}}|\boldsymbol{h})$ | Image Encoder | $E_o(\boldsymbol{x}|\boldsymbol{o})$ : pre-trained ResNet18 |
| | Measurement Vector Encoder | $E_v(\boldsymbol{m}|\boldsymbol{v})$ : MLP(linear layers = 3, hidden_units = 128, output_dim = 128, activiation = ReLU($\cdot$)) |
| | (RSSM) Deterministic State Model | $f_d(\boldsymbol{c}_t|\boldsymbol{c}_{t-1}, \boldsymbol{z}_{t-1})$ : GRU (input_size = 256, hidden_size = 256) |
| | (RSSM) Stochastic State Model | $q_z(\boldsymbol{z}_t|\boldsymbol{c}_t, \boldsymbol{m}_t, \boldsymbol{x}_t)$ : (mean, std) $\sim$ MLP (linear layers = 3, hidden_units = 256 + 128, output_dim = 128 × 2, activiation = ReLU($\cdot$)) |
| | Extracted Representation | $\hat{\boldsymbol{s}}_t = \boldsymbol{c}_t \oplus \boldsymbol{z}_t$ = Tensor(shape : $[1, 128 + 256]$) |
| Additional Modules | Action Residual Predictor | $f_r(\Delta \hat{\boldsymbol{a}}_t|\hat{\boldsymbol{s}}_t)$ : MLP (linear layers = 3, hidden_units = 256, output_dim = 2, activiation = ReLU($\cdot$)) |
| | Image Decoder | $d_o(\hat{\boldsymbol{o}}_{t+1}|\hat{\boldsymbol{s}}_t)$ : 3 ConvTranspose2d layers with activiation = ReLU($\cdot$) |
| Predictor Model $J(\hat{\boldsymbol{a}}|\hat{\boldsymbol{s}})$ | Feature Mapper | $j_m$ : MLP (linear layers = 3, hidden_units = 512, output_dim = 512, activiation = ReLU($\cdot$)) |
| | Throttle/brake Classifier | $j_c$ : MLP (linear layers = 4, hidden_units = 512, output_dim = 2, activiation = ReLU($\cdot$)), and Sigmoid($\cdot$) as output transform function |
| | Action Predictor | $j_a$ : MLP(linear layers = 4, hidden_units = 512, output_dim = 2, activiation = ReLU($\cdot$)) |
| | Output Action | $\hat{\boldsymbol{a}}_t = [\hat{a}_{steer,t}, \hat{a}_{acc,t}]$ = Tensor(shape : $[1, 2]$) |

linear layer connects the computed last historical information vector $c_t$ in timestep $t$ and the sampled hidden state $z_t$, and maps them into $\hat{s}_t$, a feature vector with length = $384$.

The feature mapper $j_m$ is a MLP as (linear layers = $3$, hidden_units = $512$, output_dim = $512$, activation = $\text{ReLU}(\cdot)$), which processes features that feed into $j_c, j_a$. The throttle/brake classifier is a MLP as (linear layers = $4$, hidden_units = $512$, output_dim = $2$, activation = $\text{ReLU}(\cdot)$), with a Sigmoid($\cdot$) transform the prediction in range $(0, 1)$. The action predictor $j_a$ receives outputs from $j_m$ and $j_c$, computes its prediction through a MLP with (linear layers = $4$, hidden_units = $512$, output_dim = $2$, activiation = $\text{ReLU}(\cdot)$).

### A.2.3 TRAINING

For 3CIL and baselines we used for comparison that can conduct representation learning and policy learning in separate stages (i.e., DIGIC, PALR, Premier-TACO, RNC, and RAP in Section 4.1), we first conduct their corresponding representation learning with expert demonstrations to obtain stable representation models. For these methods, we use an Adam optimizer (Kingma, 2014) with a learning rate set to $5e\!-\!6$ to optimize their representation models. An early-stopping monitor module is also added to prevent models from overfitting the training set, with an evaluation set $\mathbf{D_v}$ divided from the training set $\mathbf{D_e}$ in a dividing factor $10\%$. After training an epoch on the training set, the representation model is required to run on $\mathbf{D_v}$. If the performance increment in $\mathbf{D_v}$ is lower than the optimization threshold $1e-3$, this optimization epoch is marked as a potential invalid update. When the consecutive invalid updates that a representation model encountered have reached the early-stopping threshold (set to $10$ in this phase), the optimization process of the representation is finished.

When the training process for representation models was finished, we fixed these models' weights and deployed them in the following policy training phase. Similar to the former stage, an Adam optimizer with a learning rate set to $1e-6$, and an early-stopping monitor with an optimization threshold of $1e-6$ and an early-stopping threshold of $20$ are used.

During the representation learning phase of (3CIL, DIGIC, PALR, Premier-TACO, RNC, and RAP) and overall optimization of CIL, data augmentation operations are applied on the imitator's image observations to increase the diversity of the dataset and enhance the robustness of methods. Added data augmentation operations and their corresponding probability are: horizontal flip with probability $= 0.3$, color jitter (brightness $= 0.4$, contrast $= 0.4$, saturation $= 0.4$ and hue $= 0.1$) with probability $= 0.4$, and gray-scale with probability $= 0.2$.

## A.3 EXPERIMENTAL DETAILS

### A.3.1 BASELINES

As described in Section 4.1, we have picked and implemented baselines including CIL, Keyframe (Wen et al., 2021), DIGIC (Chen et al., 2024), PALR (Seo et al., 2023), Premier-TACO (Zheng et al., 2024), RNC (Zha et al., 2023), and RAP (Chuang et al., 2022) in our experiments. We now introduce each baseline and our implementations.

The Conditional Imitation Learning (**CIL**) approach (Codevilla et al., 2018) eases the complex vision-based driving task by introducing conditions (i.e., the expert's intention, often expressed as route commands) to the model's input. The vanilla CIL pipeline operates in a supervised learning manner by directly minimizing the prediction difference between the policy and expert demonstrations, as: $\mathcal{L}_J = \frac{1}{N} \sum_{i=1}^{N} \mathcal{L}_a(a_t, J(o_t, v_t))$. In our implementation, we replace the input tuple $(o_t, v_t)$ with $\hat{s}_t \sim G(\hat{s}_t | h_t)$, allowing the representation to capture more information from the temporal aspect. The overall model is still optimized by only the prediction loss.

The Keyframe-Focused Visual Imitation Learning (**Keyframe**) approach (Wen et al., 2021) alleviates the copycat problem by introducing sample-weighting strategies, based on precomputed "action prediction error" (APE) between a copycat policy and the expert demonstrations. Concretely, a copycat policy $\pi_c(\hat{a} | a_{t-l:t-1})$ is trained to use only previous actions $a_{t-l:t-1}$ to predict current action $\hat{a}$. The policy $\pi_c$ is then used to locate frames that are more likely to be changepoints, by identifying the samples with high APEs. Higher weights are then assigned to these identified samples, regularizing the imitator to focus on changepoints. In our implementation, we train a copycat policy whose

structure and input data are set similar to Wen et al., and use their step($\cdot$) function to map samples' APEs into weights on samples. The samples' weights are then plugged into the learning process of CIL.

The Domain Generalizable Imitation Learning by Causal Discovery (**DIGIC**) (Chen et al., 2024) framework, may stand as a representative for approaches that combine causal discovery and IL. Specifically, they picked the covariates that directly contribute to expert action $a_t$ as the input of the downstream predictor. In our implementation, the causal discovery is operated upon the extracted features from the representation model $G(\hat{s}_i|h_i)$. The representation model is first trained with the image reconstruction and posterior regularization losses, to ensure it captures rich information from raw history observations while achieving feature compression. The causal discovery task is then conducted with the mutual information regression test which is provided by the causal discovery toolbox (Kalainathan et al., 2020). Only features that exhibit test statistics that are higher than a threshold(0.20) are picked as input for the downstream BC predictor.

The Past Action Leakage Regularization (**PALR**) method (Seo et al., 2023) bypasses the causal confusion problem with a regularization on the conditional dependence between extracted representation $\hat{s}_t$ and expert's previous action $a_{t-1}$, given current expert's action $a_t$, as $\mathcal{L}_{\text{reg}}(\hat{s}_t; a_{t-1}, a_t)$. Following their work, we adopt the Hilbert-Schmidt conditional independence criterion (HSCIC) from (Park & Muandet, 2020) to perform past action leakage regularization in a non-parametric manner, as: $\mathcal{L}_{\text{reg-HSCIC}}(\hat{s}_t; a_{t-1}, a_t) = \text{HSCIC}(s_t, a_{t-1}|a_t)$. Such regularization term is incorporated in the representation learning stage in our implementation, with parameters set as $ridge\_lambda = 1e-3$ and $reg\_coef = 0.1$.

The **Premier-TACO** framework (Zheng et al., 2024) employs a temporal action-driven contrastive loss function for visual representation pretraining, with a new negative example selecting strategy. For a state $\hat{s}_t$, its corresponding positive example is $\hat{s}_{t+k}$, while its negative examples are then selected based from a window with size w centered at state $\hat{s}_{t+k}$ within the same episode, as $\hat{s}_{t,\text{neg}} \sim (\hat{s}_{t+k-w}, ..., \hat{s}_{t+k-1}, \hat{s}_{t+k+1}, ..., \hat{s}_{t+k+w})$. We incorporate this framework in our representation learning stage with positive stride $k = 4$ and window size $w = 5$. The action encoder utilized in Premier-TACO is implemented with a three-layer MLP with 256 hidden units.

Different from Premier-TACO that constructs positive pairs and negative pairs based on temporal indexes, the Rank-N-Contrast framework (**RNC**) (Zha et al., 2023) conducts supervised contrastive learning to shape a robust representation space with guidance from samples' continuous labels. As part of optimization target in our 3CIL method, the $\mathcal{L}_{\text{RNC}}$ in Eq 3 help aligning the distances of samples in the representation space with distances in their labels. To evaluate the benefit from $\mathcal{L}_{\text{RNC}}$ solely, we set RNC as one of our baseline, with implementation as adding it alongside with the image reconstruction and posterior regularization losses.

The Residual Action Prediction (**RAP**) method (Chuang et al., 2022) aims to resolve the copycat problem by designing the residual action prediction objective Eq 2. This approach is also introduced as a baseline, without the sample-weighting strategy in Eq 4.

To alleviate the performance bias incurred by different model capacities, all baselines in our experiments share same architecture design in Appendix A.2.2 and training strategy in A.2.3. Therefore, the major differences in metrics will come from methods' designs.

### A.3.2 PLATFORMS

All models used in experiments was trained on a batch size of 64 on a workstation with a RTX4090 GPU. In the testing phase, these models are deployed to the CARLA simulator (version 0.9.12) on another workstation with a RTX3080 GPU.

Table 3 lists configurations for the CARLA simulator used in our experiments, both collecting training data and evaluating models. Specifically, the history subsample frequency is set lower than the actual interaction frequency, as the dynamics in urban driving environments do not contain many high frequency components. Similarly, the frequency of computing and reporting metrics (reward, collision) is set to 4Hz.

Table 3: Configurations of CARLA simulator in experiments.

| Configuration | Value |
|---|---|
| System platform | Windows 10 |
| Graphics quality | quality-level=Epic |
| Interaction frequency | 20Hz |
| History subsample frequency | 4Hz |
| Perception window length | $l = 4$ |
| Metrics computation frequency | 4Hz |

### A.3.3 ENVIRONMENTAL PARAMETERS

In the conducted experiments, we modified the weather condition, traffic density, and camera parameter for each scenario used in the evaluation, we listed the environmental parameters used in expert demonstrations and evaluation process in Table 4, and introduce the effect of changing these parameters as follows.

Table 4: Environmental parameters in the training set and test stage.

| Environmental parameters | Training set | Test stage |
|---|---|---|
| Towns | Scenario 1, 2, 3, 4 ( Town01, Town03, Town04, Town06) | Scenario 1, 2, 3, 4, 5, 6 ( Town01, Town03, Town04, Town06, Town02, Town05) |
| Weather group | 'ClearNoon', 'WetNoon', 'HardRainNoon', 'ClearSunset' | 'WetCloudyNoon', 'SoftRainSunset', 'WetSunset', 'HardRainSunset' |
| Number of vehicles | Scenario 1: [80, 160], Scenario 2: [40, 100], Scenario 3: [100, 200], Scenario 4: [80, 160] | Scenario 1: 120, Scenario 2: 70, Scenario 3: 200, Scenario 4: 120, Scenario 5: 70, Scenario 6: 120 |
| Front camera FOV | Scenario 1: 70, Scenario 2: 80, Scenario 3: 100, Scenario 4: 120 | Scenario 1: 75, Scenario 2: 105, Scenario 3: 95, Scenario 4: 85, Scenario 5: 90, Scenario 6: 110 |

Except for four scenarios that appeared in the training set, two new scenarios are also included in the test stage. As the $v_{route\_command}, v_{route\_command\_next}$ are offered as navigation information, driving in unfamiliar towns is less terrifying. Still, the introduced new scenarios can examine the applicability of learned patterns of each imitator in new domains.

The weather condition is set differently in the training set and test stage as shown in Table 4, no weather condition in the test stage has been introduced to the imitator in the training dataset. Although the weather in the CARLA simulator does not affect the vehicle's physics, it does affect the lighting condition and the visibility of visual-based imitators, new weather conditions impose trials on imitators' ability in generalization. Moreover, the distribution of weather conditions in the test stage is shifted: sunsets and rain conditions more frequently appear, making the driving task more difficult.

We modify the traffic density by setting the number of other vehicles. Naturally, denser traffic leads to harder challenges on imitators' strategies. In particular, we increase the vehicle count to 200 in Scenario 3, corresponding to Town04 in CARLA, which is a small town. This contradiction between compact map size and heavy traffic load poses stress on imitators and leads to their highest collision rates in the design experiments.

Finally, the camera parameter we modified is the field of view (FOV) of the RGB camera that produces the image observation for imitators. Higher FOV means an imitator can perceive information with a wider perception range, but also brings distortions to objects in an image's corner. Moreover, an object located in an identical spot will be depicted in different sizes when setting FOV at different

levels. Such variances in observation further intensify the extent of distribution shift in the test stage and pose threats to the imitators' generalization.

Figure 5 shows samples from scenarios with environmental parameters set differently.

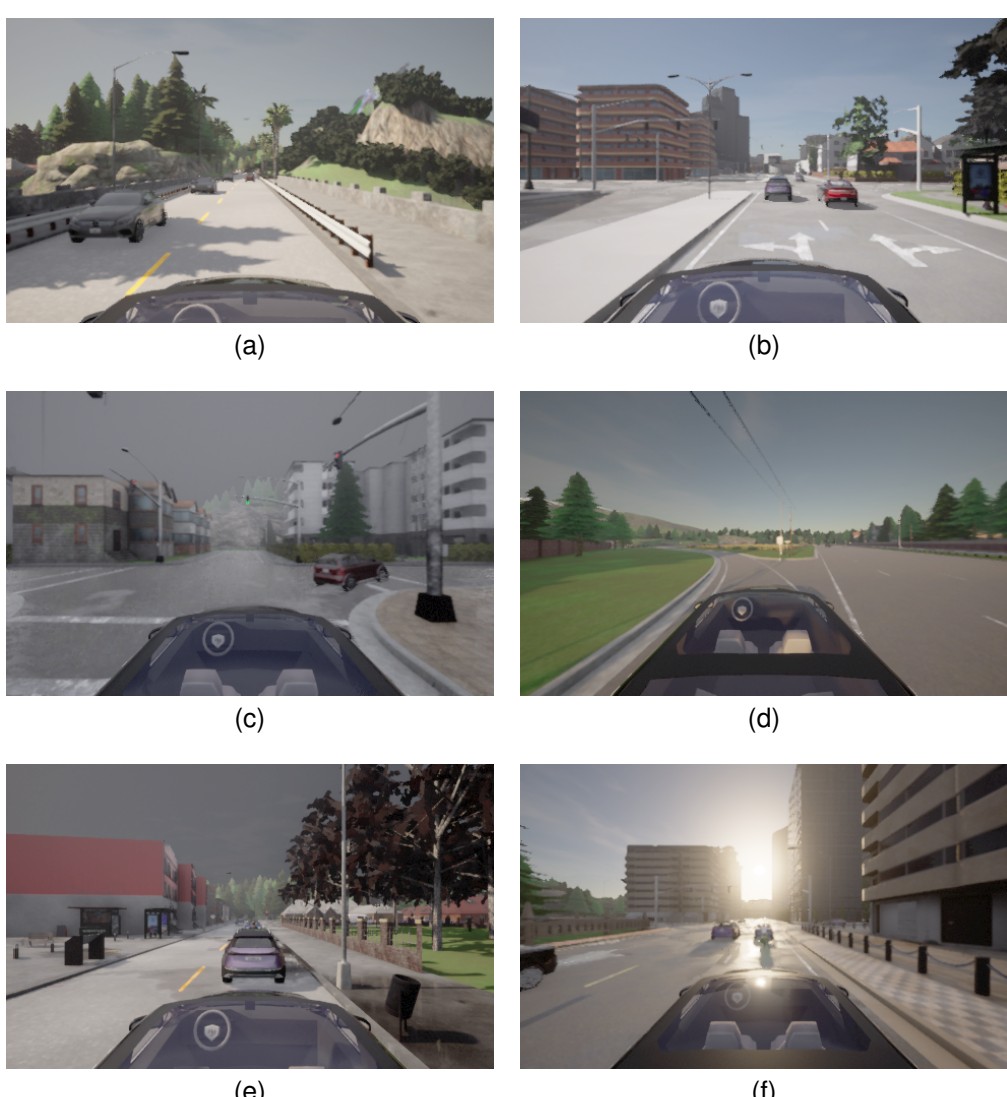

Figure 5: Illustrations of six scenarios used in our experiments. (a): Scenario 1 with the weather set to 'ClearNoon' and camera FOV set to 70. (b) Scenario 2 with the weather set to 'WetNoon' and camera FOV set to 90. (c) Scenario 3 with the weather set to 'HardRainNoon' and camera FOV set to 80.(d): Scenario 4 with the weather set to 'ClearSunset' and camera FOV set to 100. (b) Scenario 5 with the weather set to 'WetCloudyNoon' and camera FOV set to 110. (c) Scenario 6 with the weather set to 'WetSunset' and camera FOV set to 120.

### A.3.4    TEST SUITES

During evaluation, an imitator is required to drive through multiple preset routes in each scenario. A run corresponding to a route is terminated when the imitator: reaches the destination, runs out of the time limit, has a collision with other objects, or is stuck in a place for a while.

Concretely, we picked 10 routes for Scenario 1, 20 routes for Scenario 2, 20 routes for Scenario 3, 6 routes for Scenario 4, 10 routes for Scenario 5, and 10 routes for Scenario 6. In the test stage,

an imitator needs to drive in a route four times, corresponding to the set four weather conditions in Table 4. We set the run time limit to 2000 timesteps, and the stuck detection period is set to 600.

### A.3.5 REWARD DESIGN

The reward function is organized as: $R = r_{speed} + r_{position} + r_{rotation} + r_{action}$.

In a certain timestep $i$, $r_{speed}$ computes the speed reward signal based on the difference between the imitator's current speed $v_{speed,i}$ and desire speed $r_{desire\_speed}$. The desired speed $r_{desire\_speed}$ varies when the imitator is around different kinds of objects, set the maximum speed limit for imitator as $maximum\_speed = 30$, the $r_{desire\_speed}$ is computed as

$$r_{desire\_speed} = \min(maximum\_speed, maximum\_speed \times vehicle\_factor, \\ maximum\_speed \times light\_factor, maximum\_speed \times sign\_factor), \tag{5}$$

where $vehicle\_factor, light\_factor, sign\_factor$ are modified from the code of Zhang et al. (2021b). Take $vehicle\_factor$ as an example: it first locates the nearest hazard vehicle in the ego vehicle's local coordinate as $\boldsymbol{loc\_veh}$, and computes the distance $dist\_veh = \max(0, \|\boldsymbol{loc\_veh}\|_2 - b_{veh})$ with base distance $b_{veh} = 8$, then runs through a bounding function as $vehicle\_factor = \text{bound}(dist\_veh, 0, 5)/5$. $light\_factor, sign\_factor$ are computed similarly except the different base distances $b_{light} = 6$ and $b_{sign} = 5$. The speed reward is then computed as:

$$r_{speed} = 1 - \frac{|v_{speed,i} - r_{desire\_speed}|}{maximum\_speed}. \tag{6}$$

The position signal $r_{position}$ is computed based on the imitator's lateral distance $d_{lateral}$ with navigation point: $r_{position} = -1 \times (d_{lateral}/2)$.

The rotation punishment $r_{rotation}$ is computed based on the rotation yaw angle differences between the imitator and the navigation point $d_{yaw}$, as: $r_{rotation} = -1 \times \text{deg2rad}(\text{abs}((d_{yaw} + 180)\%360 - 180))$, where $\text{deg2rad}(\cdot)$ is the function converts angles from degrees to radians.

The $r_{action}$ signal punishes the imitator with $r_{action} = -0.1$ if $|a_{steer,t} - a_{steer,t-1}| > 0.01$ else $r_{action} = 0$.

## A.4 ADDITIONAL RESULTS

In this section, we present an empirical test on assumptions made by 3CIL, and a further ablation study on the utilities of each module in 3CIL. We select three representative scenarios from Table4 to conduct experiments on: Scenario 1, Scenario 5, and Scenario 6.

### A.4.1 EFFECTS OF HISTORY

An assumption adopted by our work, and previous works that belong to Behavioral Cloning from Observation Histories (BCOH) or POMDP-related approaches is: that using only the most recent frame $\boldsymbol{o}_t$ cannot provide enough essential information for agents to recover an expected policy. Therefore, it is common to design policies that utilize observation history, such as expanding the temporal perceived range of models and using suitable networks for capturing temporal dependency.

To verify whether history helped these approaches capture more crucial information, we designed an intervention analysis similar to (Chuang et al., 2022). Specifically, we replace the original history $\boldsymbol{h}_t = (\boldsymbol{o}_{t-l:t}, \boldsymbol{v}_{t-l:t})$ with the counterfactual history: $[\text{repeat}((\boldsymbol{o}_t, \boldsymbol{v}_t), l)]$, which is replacing all frames in the history with current frame $(\boldsymbol{o}_t, \boldsymbol{v}_t)$. Figure 6 provides examples of the factual setting and the counterfactual history setting. The average performances of models that deployed in experiments under counterfactual history setting are recorded, and compared to their performance under the setting of original history.

Therefore, the difference between performances in these two settings can be seen as the effect of incorporating observation history. We report the results of three methods (CIL, Premier_TACO) in three scenarios, as Table 5. Clearly, for all methods, replacing the original history with the counterfactual history will incur performance degeneration in most Scenarios. Such a phenomenon

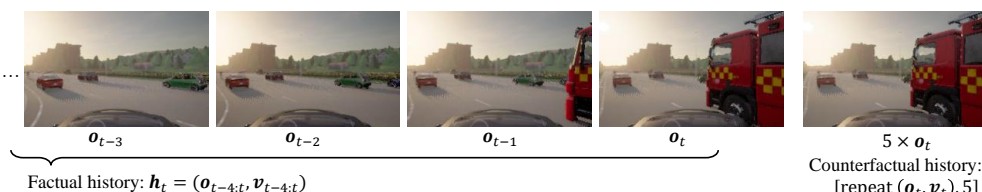

$$\underbrace{\quad}_{\text{Factual history: } \boldsymbol{h}_t = (\boldsymbol{o}_{t-4:t}, \boldsymbol{v}_{t-4:t})}$$

Figure 6: An illustration of the factual history setting and the counterfactual history setting.

Table 5: The extent of degeneration in performance when switching to the counterfactual history setting. The ratios of reward dropping and collision rate increasing are computed with the methods' performance in the factual setting (Table 1).

| Metric | Method | CIL | Premier_TACO | 3CIL |
|---|---|---|---|---|
| Scenario 1 | Dropped R (%)↓ | 35.09 ($331 \to 215$) | 13.64 ($470 \to \mathbf{406}$) | 28.82 ($521 \to 371$) |
| | Increased C (%)↓ | 12.12 ($0.66 \to 0.74$) | -11.76 ($0.85 \to 0.75$) | 29.63 ($0.54 \to \mathbf{0.70}$) |
| Scenario 5 | Dropped R (%)↓ | 685.51($7 \to -42$) | 33.01 ($517 \to 346$) | 26.44 ($539 \to \mathbf{396}$) |
| | Increased C (%)↓ | 20.68 ($0.29 \to \mathbf{0.35}$) | 59.46 ($0.37 \to 0.59$) | 22.92 ($0.48 \to 0.59$) |
| Scenario 6 | Dropped R (%)↓ | 220.53 ($46 \to -55$) | 38.95 ($331 \to 203$) | 30.34 ($447 \to \mathbf{312}$) |
| | Increased C (%)↓ | 61.76($0.34 \to \mathbf{0.55}$) | 35.29 ($0.68 \to 0.92$) | 16.95($0.59 \to 0.69$) |

suggests that models can learn patterns from transitions in observations, and observations in the past will contribute to the prediction quality.

Besides, as proposed in (Chuang et al., 2022), the performance under the counterfactual history setting can also reflect the models' capability in severe copycat status: frozen observations suggest the vehicle is in the stationary state, introduce more evident spurious correlations between current action $\hat{a}_t$ and previous actions $\hat{a}_{t-n:t-1}$. With this insight, we further investigate the degeneration extent of each method.

Interestingly, the ranks of performance under the counterfactual setting (pointed by $\to$) of methods: CIL > 3CIL > Premier_TACO in collision rate, 3CIL > Premier_TACO > CIL in reward, are roughly aligned with their original performance ranks in Table 1. Although the fixed observations in history prevent the models from inferring further information as well as introduce severe causal confusion, still our 3CIL approach manages to achieve relatively less degeneration. This may attributed to the sample-weighting strategy which is utilized in the policy learning phase of 3CIL: as the representation model failed to capture the variations within history from the frozen observations, such a deviation from learned patterns is akin to the samples with high $\boldsymbol{weight}_t$ due to failures in action residual prediction, which belongs to the circumstances we emphasis the 3CIL model to learn.

### A.4.2 SAMPLE-WEIGHTING STRATEGY

To showcase the process of our proposed sample-weighting strategy in 3CIL, we pick a circumstance where a sample's $\boldsymbol{weight}_t$ is computed with high value, as shown in Figure 7. The high deviation in action residual prediction is then translated to emphasis on learning this sample.

Previous work also incorporates sample-weighting process, such as (Wen et al., 2021) used two distinct functions to map the APE into samples' weights: step($\cdot$) and softmax($\cdot$). The step($\cdot$) sorts those samples whose APEs are greater than a large proportion of overall samples' APEs (the top $10\%$ samples measured in APE), and assigns these samples with a constant weight $W$ (set to $5.0$ in their experiments), other samples are assigned with weight $1.0$. Another implementation computes the samples' corresponding weights within a batch, by feeding their APEs into a softmax($\cdot$) function.

To evaluate the effectiveness of each sample-weighting strategy, we conduct experiments by changing the weighting process in 3CIL, which results in 3 different performance statistics in Table 6. Specifically, we replace the measurement of error in (Wen et al., 2021) (APE) with the error in

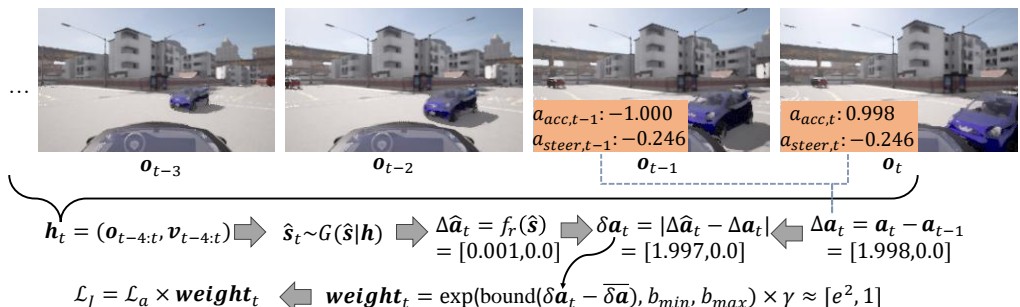

Figure 7: An illustration of the sample with high $weight_t$. For concision, $o_{t-4}$ and $v_{t-4:t}$ are omitted here. Based on features $\hat{s}_t$ obtained from the observation history $h_t$, the action residual predictor $f_r$ gives its prediction as $\Delta\hat{a}_t = [\Delta\hat{a}_{acc,t} = 0.002, \Delta\hat{a}_{steer,t} = 0.0]$ since it might be safer to remain still, given that the blue car is getting closer. However, the expert chose to accelerate to finish its left turn and give space for the upcoming blue vehicle, which led the actual residual to be: $\Delta a_t = [\Delta a_{acc,t} = 0.001, \Delta a_{steer,t} = 0.0]$. This huge disagreement $\delta a_t = [1.997, 0.0]$ results in a high sample $weight_t$, urging the predictor model to focus more on such an abnormal scene.

Table 6: The extent of degeneration in performance when replacing the sample-weighting strategy in 3CIL with None (no applying weights), step(·) and softmax(·). The ratios of reward dropping and collision rate increasing are computed with the original performance of 3CIL ( Table 1).

| Strategy / Metric | | None | step(proportion = 20%, weight = 3.0) | softmax( temperature = 0.2) |
|---|---|---|---|---|
| Scenario 1 | R ↑ | **491.77** (5.66% ↓) | 446.47 (14.3% ↓) | 389.97 (25.19% ↓) |
| | C ↓ | 0.61 (12.96% ↓) | 0.59 (9.26% ↓) | **0.57** (5.56% ↓) |
| Scenario 5 | R ↑ | **460.35** (14.51% ↓) | 338.75 (37.09% ↓) | 402.22 (25.31% ↓) |
| | C ↓ | **0.52** (8.33% ↓) | **0.52** (8.33% ↓) | 0.54 (12.50% ↓) |
| Scenario 6 | R ↑ | **401.51** (10.22% ↓) | 359.62 (19.59% ↓) | 383.05 (14.35% ↓) |
| | C ↓ | 0.71 (20.33% ↓) | 0.68 (15.25% ↓) | **0.64** (8.47% ↓) |

residual prediction. The residual prediction receives the extracted features $\hat{s}_t$ as input and can identify more general abnormal scenes beyond the copycat problem.

Out of the blue, the incorporation of sample-weighting processes does not always come with benefit (compared to "None", in the measurement of Reward). This may be caused by the potential mismatch between the functions' hyper-parameters and training data distribution, as these functions are rather sensitive to the choice of hyper-parameters: the step($\cdot$) especially, which shows extremely low average speed (3.55 in Scenario 1) when set hyper-parameters as default. We expect the performance of these sample-weighting strategies can be further improve when finetuning them with domain knowledge. Still, both step($\cdot$) and softmax($\cdot$) generally reduce the collision rate of the imitator, which may be attributed to the up-weighting on potential changepoints.

In conclusion, the choice of sample-weighting strategy is flexible, as long as the function that computes weights is a monotonic non-decreasing function of the action residual prediction error, similar to the setting of (Wen et al., 2021). However, it is essential to tune the function to achieve a balance between overly flat (not enough emphasis on abnormal scenes) and overly steep (potentially underfitting with ordinary driving scenes).

### A.4.3 EFFECTIVENESS OF EACH DESIGN

Table 7: Ablation studies.

| Method / Metric | | No $\mathcal{L}_{ar}$, no $\boldsymbol{weight}_t$ | No $\mathcal{L}_{RNC}$, no $\boldsymbol{weight}_t$ | No $\mathcal{L}_{RNC}$ | No $\boldsymbol{weight}_t$ | 3CIL |
|---|---|---|---|---|---|---|
| Scenario 1 | R ↑ | 411.31 | 383.54 | 476.35 | 491.77 | **521.26** |
| | C ↓ | 0.55 | 0.60 | 0.57 | 0.61 | **0.54** |
| Scenario 5 | R ↑ | 299.72 | 302.50 | 387.24 | 460.35 | **538.50** |
| | C ↓ | 0.50 | 0.59 | 0.55 | 0.52 | **0.48** |
| Scenario 6 | R ↑ | **447.44** | 195.53 | 234.44 | 401.51 | 447.24 |
| | C ↓ | 0.64 | 0.63 | 0.63 | 0.71 | **0.59** |

We examine the effect of each design decision in our approach and present the results in Table 7.

The results show that all major designs in 3CIL have contributed to the overall performance. Concretely, the supervised contrastive learning loss $\mathcal{L}_{RNC}$ ("No $\mathcal{L}_{ar}$, no $\boldsymbol{weight}_t$") provides important guidance in arranging the representation space, helps the imitator to achieve better alignment with the expert policy. While the presence of action residual prediction task $\mathcal{L}_{ar}$ enhances the imitator's capability in capturing crucial influence from previous actions $\boldsymbol{a}_{t-1}$, the $\mathcal{L}_{ar}$ stands alone ("No $\mathcal{L}_{RNC}$, no $\boldsymbol{weight}_t$") does not provide satisfactory gains. After incorporating the weighting strategy, the performance of an imitator ("No $\mathcal{L}_{RNC}$") does show significant improvement, but still fails to generalize in some cases, which emphasizes the importance of contrastive learning in shaping a robust representation space. Finally, the divergence between "No $\boldsymbol{weight}_t$" and 3CIL provides the evidence that adjusting weights on diverse samples is beneficial, as it can guide the imitator to focus on crucial changepoints and abnormal scenes.

