# OpenReview forum: "3CIL: Causality-Inspired Contrastive Conditional Imitation Learning for Autonomous Driving"
_ICLR.cc/2025/Conference — Submitted to ICLR 2025_

### Official Review · Reviewer_9A5c · 2024-10-31

**Soundness:** 2
**Presentation:** 2
**Contribution:** 3
**Rating:** 5
**Confidence:** 5

**Summary:**

The authors propose to combine causal reasoning techniques to assist imitation for autonomous driving. Specifically, the paper presents a novel approach, causality-inspired contrastive conditional imitation learning (3CIL), which integrates contrastive learning and action residual prediction. The framework is based on POMDP, trying to mimic the scenarios when the expert and imitator share different views.

**Strengths:**

The idea of combining causality, contrastive learning and conditional imitation learning is quite interesting. The performance seems to be good for all scenarios. Fig. 1 is intuitive.

**Weaknesses:**

My major concerns lie in theoretical clarity, experimental design, and practical applicability. In the absence of strong theoretical guarantees, the paper would benefit greatly from robust experimental results. Additionally, it’s crucial to provide a clear rationale for the integration of causality, contrastive learning, and conditional imitation learning, explaining why this combination is necessary. I have outlined specific questions below.

**Questions:**

### **Some high-level questions**
1. In "3CIL: Causality-Inspired Contrastive Conditional Imitation Learning," the abbreviation "CIL" is first defined in line 162. However, it’s unclear what the "C" stands for. Is it "Causality" or "Conditional"? Could you clarify this term for consistency?

2. Does the input for the VAE include all the past history observations $o_{1:t}$?

3. Some important details are in the appendix. The authors should consider move them to the major context, e.g., Fig. 4 and Fig. 5.

### **Some questions about theoretical guarantee**
1. As discussed in (Ruan et al., 2022; Ruan & Di, 2022; Kumor et al., 2021), unobserved confounders (UCs) often complicate causal identifiability, and certain variables must be considered to mitigate the influence of spurious correlations or shortcuts during the learning process. In your approach, which specific variables are crucial to achieving this objective? E.g., which variables are required to block all the backdoor paths.

2. Following up on the previous question, if all the past ground-truth states $s_{1:t}$ are unobserved (common in POMDPs), and $a_{t-1} \leftarrow s_{t-1} \rightarrow s_{t} \rightarrow a_{t}$ is active, in other words, there is an active backdoor path between $a_{t-1}$ and $a_{t}$ which can never be blocked. Given that this path cannot be blocked, is the policy learning process identifiable or robust? How do you ensure convergence in policy learning under these circumstances?

3. The idea of training a representation model $G$ is not that novel. Especially, when $\hat{s}_{t}$ is unobserved, it is very hard to directly determine whether the representation model is good or not. While simulations may provide insights, evaluating the model’s practical effectiveness in real-world conditions can be significantly harder. What methods do you suggest to compare model performance outside of a simulation environment?

4. The proposed method appears to be a pipeline structure (i.e., representation model + policy model).

     4.1 IIf overall performance is not expected, what strategy would you recommend to isolate and improve the specific component responsible? How can one effectively determine whether limitations stem from the representation model or from the policy?

     4.2 Will there be any cascaded errors from upstream to downstream tasks? Could you elaborate on any mechanisms in place to mitigate such cascading errors?

### **Some questions about experiments**

1. In your experiments, does the imitator have access to the reward $R$? Additionally, are the expert demonstrations generated from RL algorithms that use the same reward function $R$?

2. For the observations, the expert is able to observe $s_{t}$, but the imitator is only able to observe $o_{t}$. Is that correct?

3. To what extent does the reward $R$ reflect real-world driving behaviors? How accurately does it capture the dynamics observed in actual driving scenarios?

4. Could the authors add more details to the reward $R$? Specifically, how are the four components $r$ defined? If the primary objective is to evaluate route adherence, is $r_{position}$ alone sufficient, or are the other rewards essential? Please clarify the role of each reward component.

5. Additional metrics could enhance the evaluation process, such as the RMSE between predicted positions and target routes. Would the authors consider including these metrics for a more comprehensive analysis?

**Should the authors address these questions thoroughly, I would consider raising my evaluation score.**

---------

References:
- Pearl, Judea. Causality. Cambridge university press, 2009.
- Peters, Jonas, Dominik Janzing, and Bernhard Schölkopf. Elements of causal inference: foundations and learning algorithms. The MIT Press, 2017.
- Ruan, Kangrui, and Xuan Di. "Learning human driving behaviors with sequential causal imitation learning." Proceedings of the AAAI Conference on Artificial Intelligence. Vol. 36. No. 4. 2022.
- Ruan, Kangrui, et al. "Causal imitation learning via inverse reinforcement learning." The Eleventh International Conference on Learning Representations. 2023.
- Kumor, Daniel, Junzhe Zhang, and Elias Bareinboim. "Sequential causal imitation learning with unobserved confounders." Advances in Neural Information Processing Systems 34 (2021): 14669-14680.

---

> ### Author Response · Authors · 2024-11-25
> **Response to Reviewer 9A5c 1/3**
>
> We thank the reviewer for the constructive comments and positive feedback on our paper. Regarding the concerns of the Reviewer 9A5c, we provide the following response.
> >**Q1.1 What the "C" in CIL stands for.**
>
> Sorry for this confusion. The "CIL" stands for "Conditional Imitation Learning", and the term "3CIL" represents **C**ausality-Inspired **C**ontrastive **C**onditional **I**mitation **L**earning. We have corrected such abbreviation confusion in the revised manuscript.
>
> >**Q1.2. Does the input for the VAE include all the past history observations $o\_{1:t}$ ?**
>
> Sorry for this confusion. The input for the VAE contains only history observation within the time window $l$, as $o\_{t-l:t}$, $l$ is set to $5$ in our experiment. We have added Table 2 to introduce the structures of data and components in our approach, in Appendix A.2.2.
>
> >**Q1.3. Some important details are in the appendix.**
>
> Thanks for the valuable feedback. Due to the page limit of the conference, we have to move some important parts of our paper to the appendix. We have added refer links in the main context to help locate corresponding contents, and we will keep working on improving the presentation of our paper.
>
> >**Q2.1. Remedies for unobserved confounders(UCs).**
>
> We agree with the reviewer that UCs played vital roles in causal inference and imitation learning, as they hindered the identification of causal effects, and introduced spurious correlations in imitators' decision process.
> However, different from previous approaches that provide feasible procedures for adversarial imitation learning (AIL)[1], inverse reinforcement learning (IRL) [2], and graphical criterion [3,4] for deriving imitators that match or outperforms the expert's performance, our work majorly focuses on assisting the complex vision imitation learning task with the setting of behavior cloning (BC).
> Therefore, our work is more similar to the settings of [5,6], where the high-dimensional observation prevents the imitator from utilizing specific variables to achieve provably de-confounding.  However, the imitator may benefit from the incorporation of additional supervision signals or features that exhibit correlations to UCs and observations, so that the backdoor paths can be blocked by controlling them.  For example, adding neighbors' actions as instrumental variables to control the confounding, as they affect $a\_t$ through $s\_t$, and have a strong correlation with $s\_t$.
>
> >**Q2.2. Active backdoor path $a\_{t-1}\gets s\_{t-1}\to s\_t\to a\_t$ never be blocked.**
>
> We appreciate the reviewer’s insightful questions. As pointed out, the unobservability of the ground truth states $s\_{1:t}$ introduces inevitable confounding factors in the imitator's decision-making process. Without the introduction of additional features, more interaction chances with the environment during the training phase, or further domain knowledge, the active backdoor path $a\_{t-1}\gets s\_{t-1}\to s\_t\to a\_t$ cannot be blocked in this setting. As a result, providing guarantees for identifiability or robustness is not feasible within the scope of our current framework.
> Regarding convergence, while we do not offer formal guarantees under these circumstances, we have incorporated supervised contrastive learning and a sample-weighting strategy to assist the learning process and mitigate potential errors arising from the representation learning stage. These approaches help ensure more robust performance in practice, although formal convergence analysis remains an open direction for future work.

---

> ### Author Response · Authors · 2024-11-25
> **Response to Reviewer 9A5c 2/3**
>
> >**Q2.3.Novelty in representation model, determining the quality of representation model, gap between simulation and real environment.**
>
> We thank the reviewer for the constructive comments. Training the representation model $G$ for the imitator is indeed not fresh in the community, still, $G$ is an essential part that affects the overall imitation performance, and our contribution majorly lies in the incorporation of supervised contrastive learning which provides important guidance in the representation learning phase. As one cannot access the ground truth state $s\_t$, it is indeed difficult to determine the quality of learned representation. Therefore, we assume the optimality of the expert's action $\hat{a}\_t$, and regularize $G$ to build an alignment between inferred state $\hat{s}\_t$ and $\hat{a}\_t$ with the supervised contrastive learning loss.
> We agree that the gap of sim2real requires effort, and our current approach is far from real-world deployments. Human-in-the-loop evaluation could provide useful insights in practice. However, as the driving task in the real world requires orchestrating numerous modules, we cannot conclude that a certain evaluation will fully validate the model's performance.
>
> >**Q2.4.1. If overall performance is not expected, what strategy would you recommend to isolate and improve the specific component responsible? How can one effectively determine whether limitations stem from the representation model or from the policy?**
>
> We thank the reviewer for the valuable questions. The separation of learning representation model $G$ and learning policy $J$ does bring complexities in optimizing toward the driving task.
> In our work, as we introduce the future image observation reconstruction task $\mathcal{L}_{\text{fo}}(\hat{o}\_{t+1}, o\_{t+1})$ to provide supervisions on $G$'s capability in inferring dynamics and preserving information of raw observations $h\_{t} = (o\_{t-l:t}, v\_{t-l:t})$, the quality of the reconstructed image can provide clues for analyzing performance limitations on the representation model side. We can also permute the partial input of $G$ to investigate the effects of certain features on the specific components.
> On the policy side, beyond the supervision from offline datasets, we can also inspect the output smoothness, or perform minor perturbation on the obtained representation $\hat{s}\_t$ to investigate the robustness of policy.
>
> >**Q2.4.2. About cascading errors.**
>
> Thanks for the valuable questions. The design of separation will introduce errors from $G$ to $J$, as UCs remain uncontrolled, and the unobservability of the ground truth states. To mitigate such an issue, we propose a sample-weighting strategy, by utilizing the errors in the residual action prediction (RAP) task as proxies of the extent of error in $G$. We assign high weights on samples that obtain high errors in RAP, to enforce the policy $J$ learning such abnormal scenes. With this mechanism, $J$ is encouraged to match with the expert behaviors in these samples properly, even though the upstream $G$ has failed to do so. A detailed discussion and the visualization of the sample-weighting process are provided in Appendix A.4.2.
>
> >**Q3.1. Imitator's access to Reward $R$, same reward fucntion for the expert?**
>
> Thanks for the valuable questions. In our experiments, the imitator has no access to $R$, both in the training and testing stages. The expert demonstrations are generated by an RL agent which is pre-trained with PPO, and with the same reward function.
>
> >**Q3.2. For the observations, the expert is able to observe $s\_t$, but the imitator is only able to observe $o\_T$. Is that correct?**
>
> That is correct. In our experiments, the expert has access to the ground truth information, in the form of bird-eye view images, $s\_t$ stands as the information within the bev images in timestep $t$. The imitator only has access to $o\_t$, the image captured by the RGB camera, and a corresponding measurement vector $v\_t$ which provides navigation commands and the speed of the ego vehicle.

---

> > ### Author Response · Authors · 2024-11-25
> > **Response to Reviewer 9A5c 3/3**
> >
> > >**Q3.3 & Q3.4. About reward function.**
> >
> > Thank you for your valuable questions. We have provided a detailed introduction about the components of $R$ in Appendix A.3.5. Here, we briefly state the purposes of components in the reward $R=r_{speed} + r_{position} + r_{rotation}+r_{action}$.
> > Based on the current circumstance of the agent (e.g., has another vehicle close to itself, or in front of a red light), the desired speed of the agent is varying, which affects the assignment of  $r_{speed}$. The two components $r_{position}$ and $r_{rotation}$ are computed based on the distance and angle difference with respect to the nearest navigation point. The $r_{action}$ punishes the agent for large variations in its actions.
> > Due to the complexity of real-world driving behavior, the $R$ used in our work is not enough to fully depict the desired driving policy. However, it can reflect the driving behavior to some extent, as the four components together require an agent to drive toward the destination while maintaining the corresponding desired speed for each circumstance, and avoiding steep changes in its actions. Therefore, we believe that components in $R$ are essential to evaluate an agent's performance.
> >
> > >**Q.3.5 Adding additional metrics.**
> >
> > We thank the reviewer for the constructive suggestions. We agree that additional metrics will aid the experiments and analysis. However, due to time constraints, we are unable to report the performance of methods in these metrics. We appreciate the reviewer’s suggestion, and we plan to include more metrics in future work to further validate the effectiveness of methods.
> >
> > We are sincerely grateful for your time and efforts in the review process.
> >
> > [1]Ruan, Kangrui, and Xuan Di. "Learning human driving behaviors with sequential causal imitation learning." *Proceedings of the AAAI Conference on Artificial Intelligence.*  2022.
> > [2]Ruan, Kangrui, et al. "Causal imitation learning via inverse reinforcement learning." *The Eleventh International Conference on Learning Representations.* 2023.
> > [3]Kumor, Daniel, Junzhe Zhang, and Elias Bareinboim. "Sequential causal imitation learning with unobserved confounders." *Proceedings of the 35th International Conference on Neural Information Processing Systems.* 2021.
> > [4]Zhang, Junzhe, Daniel Kumor, and Elias Bareinboim. "Causal imitation learning with unobserved confounders." *Proceedings of the 34th International Conference on Neural Information Processing Systems.* 2020.
> > [5]Seo, Seokin, et al. "Regularized behavior cloning for blocking the leakage of past action information." *Proceedings of the 37th International Conference on Neural Information Processing Systems.* 2023.
> > [6]Park, Jongjin, et al. "Object-aware regularization for addressing causal confusion in imitation learning." *Proceedings of the 35th International Conference on Neural Information Processing Systems.* 2021.

---

> ### Comment · Reviewer_9A5c · 2024-11-27
> **Officie Comment by Reviewer 9A5c**
>
> Thank you for your detailed response, which has addressed most of my concerns. To further streamline the review process and ensure transparency, I suggest that the authors consider including a concise **global response or summary of rebuttals**. This could highlight the key modifications made to the paper, specifying what is newly added and where these additions have been incorporated. While this may not be a standard practice, it would greatly enhance clarity and assist reviewers in assessing the revisions comprehensively.

---

> > ### Author Response · Authors · 2024-11-27
> >
> > Thank you for the valuable suggestion. We are glad to have addressed your concerns. We have added a general response to introduce the key modifications. We also highlighted them with blue text color in our updated manuscript. Your support would be instrumental in enhancing the quality and impact of our work.

---

### Official Review · Reviewer_MYqZ · 2024-11-02

**Soundness:** 4
**Presentation:** 3
**Contribution:** 3
**Rating:** 6
**Confidence:** 4

**Summary:**

To deal with the causal confusion problem in imitation learning, authors take the inspiration from causal learning and propose the 3CIL framework which integrates contrastive learning, action residual prediction, and importance weighting techniques together. Testing on autonomous driving benchmark CARLA, 3CIL achieves good performance with higher success rate and lower collision times than the baselines.

**Strengths:**

The proposed framework is well-motivated.

The experimental result looks good.

**Weaknesses:**

1.	For importance weighting, would you please intuitively or theoretically explain the motivation for using the errors of action residual prediction as weights, instead of other choices, e.g., AdaBoost or Keyframe? Or would you please compare with them?

2.	Why is it necessary to divide the imitating process into two separate stages? I suggest comparing with training the representation modules and the policy end-to-end.

3.	In Table 1, why is the performance of 3CIL in scenario 3 worst across all scenarios? Even worse than the unseen ones, i.e., 5 and 6.

4.	More ablation studies, analysis experiments, and visualizations are necessary. The current experimental results only contain the comparison with baselines and two simple ablation studies. I suggest having more experiments and visualizations to verify your arguments that 3CIL successfully removes the spurious correlation and importance weighting correctly finds the rare scenarios.

5.	Missing some important references [2,3,4].


[1] Keyframe-focused visual imitation learning

[2] Chauffeurnet: Learning to drive by imitating the best and synthesizing the worst

[3] Fighting Fire with Fire: Avoiding DNN Shortcuts through Priming

[4] Shaking the foundations: delusions in sequence models for interaction and control

**Questions:**

Please refer to the Weaknesses part.

---

> ### Author Response · Authors · 2024-11-25
> **Response to Reviewer MYqZ 1/2**
>
> We thank the reviewer for the constructive comments and positive feedback on our paper. Regarding the concerns of the Reviewer MYqZ, we provide the following response.
>
> >**W1. Choice of importantce weighting strategies.**
>
> Thanks for the valuable suggestion. Akin to Keyframe[1], the sample-weighting strategy in our approach aims to identify potential changepoints and abnormal scenes in the training dataset.
> Instead of using prediction error from a copycat policy in [1], we utilize the errors of action residual prediction as evidence to assign samples with different importance. Because the action residual predictor receives features from the representation model directly, which can locate important samples more precisely, compared to the copycat policy that only takes actions in previous frames as input.
> We have added the Keyframe approach as a baseline and compared it in Section 4.  We briefly report the average collision rates below.
>
> |          | Scenario 1 | Scenario 2 | Scenario 3 | Scenario 4 | Scenario 5 | Scenario 6 |
> |----------|------------|------------|------------|------------|------------|------------|
> | Keyframe[1] | 0.58       | 0.57       | 1.56       | 0.31       | 0.53       | 0.64       |
> | Ours     | **0.54**       | **0.46**       | **1.25**       | **0.27**       | **0.48**       | **0.59**       |
>
> We also added Appendix A.4.2 to further study the importance weighting process, in which both visualization and the effect of switching the weighting function are provided.
>
> >**W2. Imitating in two separate stages versus end-to-end.**
>
> We thank the reviewer for the valuable question and suggestion. We divide the learning process into two separate stages, based on the purpose of addressing the inherent complexity of the visual imitation learning task. The separation enables each module (the representation model and predictor model) to focus on distinct aspects of the task, and contributes to a more stable and efficient training process by decoupling the challenges of feature extraction and policy optimization.
> In terms of comparison with end-to-end approaches, we have implemented CIL[2] and Keyframe[1] with the end-to-end setting, we briefly report the average accumulated rewards below, alongside the results of two approaches that conduct the learning process in a two-stage manner. While the separate stages design generally works better in our conducted experiments, we look forward to further exploring the difference between separate stages and end-to-end training in our future work.
> |              | Scenario 1 | Scenario 2 | Scenario 3 | Scenario 4 | Scenario 5 | Scenario 6 |
> |--------------|------------|------------|------------|------------|------------|------------|
> | CIL[2]          | 330.49     | 12.14      | 247.29     | 345.00     | 7.18       | 45.93      |
> | Keyframe[1]     | 353.83     | 309.70     | 125.30     | 529.68     | 278.95     | 215.77     |
> | Premier-TACO[3] | 469.99    | 431.22     | 204.38     | 561.13     | 516.70     | 331.29     |
> | Ours         | **521.26**     | **587.44**     | **420.38**     | **966.35**     | **538.50**     | **447.27**     |

---

> ### Author Response · Authors · 2024-11-25
> **Response to Reviewer MYqZ 2/2**
>
> >**W3. Performance gap between Scenario 3 and other scenarios.**
>
> We appreciate the reviewer for the great efforts in reviewing. Such a gap majorly comes from the environmental parameter setting, as we set Scenario 3 to have more vehicles (=200) compared to other scenarios (mean=100). The contradiction between compact map size and heavy traffic load in Scenario 3 poses stress on imitators and leads to their highest collision rates in the design experiments.
> Moreover, the majority of methods did not exhibit significant drops in unseen environments (Scenario 5,6), which can be attributed to the conditional imitation learning setting we considered in this paper. The direct navigation information shown to the imitators can offer strong indications in the deployment phase, which eases the challenge coming from unseen environments.
>
> >**W4. More ablation studies, analysis experiments, and visualizations are necessary.**
>
> Thanks for the constructive suggestions. We have presented more studies in Appendix A.4. We briefly conclude the revisions as follows.
> (1)For more ablation studies and analysis experiments: We study the effectiveness of major components in our design framework and present them in Appendix A.4.2 and A.4.3.
> (2)For more visualizations: We present Figure 7 in Appendix A.4.2 to illustrate the process of our proposed importance weighting strategy.
> (3)For further investigate the capability of our method in the presence of severe spurious correlations, we conduct an experiment under the counterfactual history setting and present the result and discussion in Appendix A.4.1.
>
> >**W5. Missing some important references.**
>
> Thanks for pointing out the missing references. We agree that these works are truly important for imitation learning research, and we have incorporated them in the revised manuscript. Thank you again for your valuable feedback.
>
> [1]Wen, Chuan, et al. "Keyframe-focused visual imitation learning." *arXiv preprint arXiv:2106.06452.* 2021.
> [2]Codevilla, Felipe, et al. "End-to-end driving via conditional imitation learning." 2018 IEEE international conference on robotics and automation (ICRA). IEEE, 2018.
> [3]Zheng, Ruijie, et al. "Premier-TACO is a Few-Shot Policy Learner: Pretraining Multitask Representation via Temporal Action-Driven Contrastive Loss." *Forty-first International Conference on Machine Learning.* 2024.

---

> ### Comment · Reviewer_MYqZ · 2024-12-02
>
> Thanks for your response and the new results during the rebuttal period. Most of my concerns have been addressed. I will keep my score.

---

### Official Review · Reviewer_USMw · 2024-11-07

**Soundness:** 3
**Presentation:** 3
**Contribution:** 3
**Rating:** 6
**Confidence:** 4

**Summary:**

This paper proposes to solve causal confusion problem in imitation learning using supervised contrastive learning, residual prediction and sample weighting. It draws insights from causality that motivates to learn a representation of history observations without spurious correlations.
Experiments on CARLA shows solid improvements over baselines, such as CIL and Premier-TACO.

**Strengths:**

- Formulate the imitation learning problem from a causal perspective and tries to prevent confounding factors using representation learning.
- Proposes to use supervised contrastive learning to learn an image representation that aligns with expert actions.
- The improvements in the tested scenarios is promising.

**Weaknesses:**

- Lack of comparisons with some related work, such as Wen et.al. Key-frame focused visual imitation learning, which proposes a weighting strategy based on action predictability.
- It would be nice to have more quantitative and qualitative analysis of the improvement. Can we attribute those to improvement in reducing spurious correlations?
- Lack of evaluation on CARLA benchmark instead of self-constructed scenarios.

**Questions:**

The improvements in those scenarios look promising. But, how to demonstrate it's coming from reducing the confounding factors as shown in previous sections?

---

> ### Author Response · Authors · 2024-11-25
> **Response to Reviewer USMw**
>
> We thank the reviewer for the constructive comments and positive feedback on our paper. Regarding the concerns of the Reviewer USMw, we provide the following response.
>
> >**W1. Lack of comparisons with some related work, such as Key-frame focused visual imitation learning.**
>
> Thank you for this valuable suggestion. We have introduced Keyframe[1] and implemented it in our experiments, with discussions about its performance in Section 4 and Appendix A.3.1.
> We also perform a study by replacing our sample-weighting strategy with two functions in [1], a brief result is listed below, and a detailed discussion about weighting strategies is provided in Appendix A.4.2.
> |            |                | step[1] | softmax[1] | Ours |
> |------------|----------------|----------|-------------|------|
> | Scenario 1 | Collision Rate | 0.59     | 0.57        | **0.54** |
> | Scenario 5 | Collision Rate | 0.52     | 0.54        | **0.48** |
> | Scenario 6 | Collision Rate | 0.68     | 0.64        | **0.59** |
>
> >**W2. More quantitative and qualitative analysis of the improvement. Can we attribute those to improvement in reducing spurious correlations?**
>
> We appreciate the reviewer's constructive suggestions. We have added more quantitative and qualitative analysis in Appendix A.4. We also conduct an experiment under the counterfactual history setting similar to [2], which introduces more severe spurious correlations by fixing frames.
> |            |              | Premier-TACO[3] | Ours  |
> |------------|--------------|--------------|-------|
> | Scenario 1 | Reward| **405.87**      | 371.00|
> | Scenario 5 | Reward| 346.12        | **396.11** |
> | Scenario 6 | Reward| 203.43        | **311.53** |
>
> The experimental result further suggests that our approach is capable of reducing the effect of spurious correlations, and therefore, maintaining good performance.
>
>
> >**W3. Lack of evaluation on CARLA benchmark instead of self-constructed scenarios.**
>
> We appreciate the reviewer for the constructive comment. Due to the limited time and platforms, we are not able to evaluate methods on the CARLA benchmark currently.
>
> >**Question: The improvements in those scenarios look promising. But, how to demonstrate it's coming from reducing the confounding factors as shown in previous sections?**
>
> We thank the reviewer for the valuable feedback. We agree that it may not be concluded that our improvements come from controlling the confounding factors: as both the true state $s\_t$ and the inferred state $\hat{s}\_t$ are still influenced by previous actions $a\_{t-l:t-1}$. In contrast, the goal of our paper is to get rid of the reliance on shortcuts that are introduced by $a\_{t-l:t-1}$ on the imitator's decision $\hat{a}\_t$. To do this, our framework incorporates several designs to mitigate the effect of spurious correlations.
> In evaluation, we have modified several environmental parameters such as weather conditions, camera angle, and traffic density to create scenarios that are significantly different from the samples in the training set.  Therefore, the imitator is required to learn beyond the spurious correlations so that its strategy can generalize well in the testing phase, which our approach proved to do.
>
> [1]Wen, Chuan, et al. "Keyframe-focused visual imitation learning." *arXiv preprint arXiv:2106.06452.* 2021.
> [2]Chuang, Chia-Chi, et al. "Resolving copycat problems in visual imitation learning via residual action prediction." *European Conference on Computer Vision.*  2022.
> [3]Zheng, Ruijie, et al. "Premier-TACO is a Few-Shot Policy Learner: Pretraining Multitask Representation via Temporal Action-Driven Contrastive Loss." *Forty-first International Conference on Machine Learning.* 2024.

---

> > ### Comment · Reviewer_USMw · 2024-12-02
> >
> > Thank you for your response! My concerns are mostly addressed and I will keep my score.

---

### Official Review · Reviewer_31eB · 2024-11-11

**Soundness:** 2
**Presentation:** 2
**Contribution:** 2
**Rating:** 3
**Confidence:** 4

**Summary:**

The authors present a new imitation learning algorithm that aims to solve some causal confusion problems in previous imitation learning methods on self-driving tasks. Specifically, the authors propose to 1) learn a more representative state representation; 2) reduce the chance of learning spurious-correlation by inferring delta actions from latent states, and 3) weight training samples by the discrepancy between prediction and ground truth.

**Strengths:**

* The authors do a good job summarizing previous findings about causal confusion problems in self-driving tasks
* Proposed lots of interesting strategies to potentially solve or alleviate the causal confusion problems

**Weaknesses:**

* Authors made lots of assumptions:
   * It's not sufficient to directly map ot to at
     * This remains an untested hypothesis
   *  learning a decoder for \hat{s}t helps it match with expert st
      * on the contrary, learning a decoder for \hat{s}t could force the encoder to focus on every detail in the image, even the ones that do not directly contribute to ground st.
   *  Since delta(at) is inferred from (st), it doesn't learn the spurious correlation
      * This assumption can be wrong since st would contain information from a(t-1)
   * The proposed method is better than baselines in most scenarios, but is that because of the design choices or just better models or bigger capacities?
* Figure 2 could have more annotations
  *  It would be better if the authors could annotate the different colors and shapes of each node

**Questions:**

My main concern is that there are lots of assumptions made in this paper that are unsupported by evidence or experiments, I would change my opinion if the authors present more ablation experiments that carefully study each of their design decisions. I would also love to see more qualitative examples (instead of just descriptions). Lastly,  the authors should give more details when comparing the baselines. Are the performance gain simply caused by a better network architecture or a bigger network capacity?

---

> ### Author Response · Authors · 2024-11-25
> **Response to Reviewer 31eB 1/2**
>
> We thank the reviewer for the constructive comments. Regarding the concerns of the Reviewer 31eB, we provide the following response.
> >**W1.1. It's not sufficient to directly map $o\_t$ to $a_t$: This remains an untested hypothesis.**
>
> We thank the reviewer for pointing this out. Indeed, such a hypothesis may not be valid in certain circumstances. Previous studies[1,2,3] also found that the introduction of observations in the past $o\_{t-l:t-1}$ does not always comes with benefits.
> However, our paper focuses on the imitation learning scheme for the driving task.  Due to the mismatch between the imitator's observation $o\_t$ and the true states $s\_t$ utilized by the expert policy, the scenarios are commonly modeled with POMDPs[2,3].  The partial observability of actual state $s\_t$ requires the imitator to infer the true state by utilizing information in the observation history.  Therefore, we assume that it is not sufficient to directly map $o\_t$ to $a\_t$.
> We also conduct experiments to investigate the effect of history observations on the imitators' performance, by replacing $o\_{t-l:t-1}$ with values of $o\_t$. The performances of imitators have significant deteriorations, suggesting that the observations in the past did provide useful indications for imitators to recover expert policy. A detailed discussion about the effect of observation history is provided in Appendix A.4.1 in our updated manuscript.
> |            |              | Premier-TACO | Ours  |
> |------------|--------------|--------------|-------|
> | Scenario 1 | Dropped Reward (%) | 13.64        | 28.82 |
> | Scenario 5 | Dropped Reward (%) | 33.01        | 26.44 |
> | Scenario 6 | Dropped Reward (%) | 38.95        | 30.34 |
>
> >**W1.2. Learning a decoder for $\hat{s}\_t$ helps it match with expert $s\_t$: On the contrary, learning a decoder for $\hat{s}\_t$ could force the encoder to focus on every detail in the image, even the ones that do not directly contribute to ground $s\_t$.**
>
> Thank you for pointing this out. That’s true, the supervision of image reconstruction does force the encoder to focus on every detail in the image sequence, which will inevitably introduce unwanted features that are not directly related to ground $s\_t$.
> However, as we have no access to the ground truth $s\_t$, one can not distinguish all causal features (features that directly contribute to ground $s\_t$) from non-causal features without introducing further assumptions or prior knowledge. Moreover, the potential mismatch between assumptions and the underlying mechanisms of the target system will introduce risks to the overall performance of the predictors that focus on only causal features, as shown by the empirical study[4].
> Therefore, in this work, we consider maintaining information observed by the imitator as much as possible in the representation $\hat{s}\_t$, and assigning the imitator the ability to infer the future, which is examined by the quality of reconstrued future image observation. Moreover, the representation is also governed by the supervised contrastive learning loss $\mathcal{L}\_{\text{RNC}}$, encouraging it to match with expert $s\_t$, in the measurement of alignment with corresponding action label $a\_t$.
> >**W1.3. Since $\Delta a\_t$ is inferred from $s\_t$, it doesn't learn the spurious correlation: This assumption can be wrong since $s\_t$ would contain information from $a\_{t-1}$.**
>
> We totally agree with the reviewer that $s\_t$ would carry information about $a\_{t-1}$, and the influence from $a\_{t-1}$ would remain in both $\hat{s}\_t$ and $s\_t$. However, the idea of introducing the action residual prediction $\Delta a\_t$ task, is to properly express the past action influence on current action, through capturing action variations ($\Delta a\_t = a\_t-a\_{t-1}$). Therefore, we aim not to block all possible influence of $a\_{t-1}$, but to prevent the imitator from relying on shortcuts.
>
> >**W1.4. The proposed method is better than baselines in most scenarios, but is that because of the design choices or just better models or bigger capacities?**
>
> We thank the reviewer for raising this thoughtful question. To appropriately state the contribution of our approach, the baseline models that we implemented all use the same network design  (i.e., the representation model, decoder, and predictor), the same input specs, and the training strategy. Therefore, the performance differences between methods majorly lay in their design choices. We also provide detailed introductions about baselines used in our experiments and the ways we implemented them (in Appendix A.3.1 in our updated manuscript).

---

> > ### Author Response · Authors · 2024-11-25
> > **Response to Reviewer 31eB 2/2**
> >
> > >**W2. Figure 2 could have more annotations: It would be better if the authors could annotate the different colors and shapes of each node.**
> >
> > Thank you for your valuable suggestion. In the updated manuscript, we have added annotations to Figure 2.
> >
> > >**Questions about untested assumptions, more ablation experiments, more qualitative examples, details in comparing baselines, and performance gain.**
> >
> > We really appreciate the reviewer for the constructive comments. We have revised our paper based on your advice. We list the revisions as follows.
> > (1) We conduct experiments to test the assumption about the benefit of observation history and present them in Appendix A.4.1.
> > (2) We present more ablation experiments to study the effectiveness of major components in our design framework and present them in Appendix A.4.2 and A.4.3.
> > (3) We add Figure 7 in Appendix A.4.2 to provide an example of the process of our proposed framework.
> > (4) We introduce the baselines we used for comparison and describe their corresponding implementing process, and present them in Appendix A.3.1.
> > Thank you again for your valuable feedback.
> >
> > [1]de Haan, Pim, Dinesh Jayaraman, and Sergey Levine. "Causal confusion in imitation learning." *Proceedings of the 33rd International Conference on Neural Information Processing Systems.* 2019.
> > [2]Chuang, Chia-Chi, et al. "Resolving copycat problems in visual imitation learning via residual action prediction." *European Conference on Computer Vision.*  2022.
> > [3]Seo, Seokin, et al. "Regularized behavior cloning for blocking the leakage of past action information." *Proceedings of the 37th International Conference on Neural Information Processing Systems.* 2023.
> > [4]Nastl, Vivian Yvonne, and Moritz Hardt. "Do causal predictors generalize better to new domains?." *Proceedings of the 38th International Conference on Neural Information Processing Systems.* 2024.
> > [5]Chen, Yang, Yitao Liang, and Zhouchen Lin. "DIGIC: Domain Generalizable Imitation Learning by Causal Discovery." *arXiv preprint arXiv:2402.18910.* 2024.

---

> ### Author Response · Authors · 2024-12-01
> **Sorry to bother you**
>
> Dear reviewer 31eB,
>
> We really appreciate your time and efforts engaged in the review phase. As the discussion period is coming to a close, we wanted to check back to see whether you have any remaining questions or concerns. We would be happy to clarify further, and grateful for any other feedback you may provide.
>
> Thank you very much and look forward to your replies! Happy Thanksgiving Day！
>
> Best regards,
> Paper authors

---

### Author Response · Authors · 2024-11-27
**General Response**

We thank all the reviewers for their time and efforts in the review phase and the constructive comments.
**We have revised our paper (highlighted in blue text color). The modifications are summarized as follows.**
1. (For Reviewer 31eB, USMw, MYqZ). We present more quantitative analysis and experimental results, including the performance of models under severe causal confusion (Appendix.A.4.1), the effect of replacing the propose sample-weighting strategy (Appendix.A.4.2), ablation stduies on the effect of major componenets in our method (Appendix.A.4.3).
2.  (For Reviewer USMw, MYqZ).  We add the comparison with Keyframe[1], including methods' performance (Table 1 in Section 4.2), and investigation of the effect of different weighting functions (Table 6 in Appendix.A.4.2).
3. (For Reviewer 31eB, MYqZ, 9A5c). We improve the presentation of experiments, including introducing the structure of data and major components of our method (Table 2 in Appendix A.2.2), the baselines used in our experiments and the implementation details (Appendix A.3.1), our experimental platforms (Appendix A.3.2), and the definition of reward function (Appendix A.3.5).
4. (For Reviewer 31eB, USMw, MYqZ, 9A5c). We provide more visualizations, including Figure 6 in Appendix A.4.1 to illustrate circumstances with severe spurious correlations, and Figure 7 in Appendix A.4.2 to illustrate the process of sample-weighting in our method.

We appreciate the efforts of reviewers in the review and rebuttal phase, and their the valuable questions and comments,  which have materially improved our paper presentation.

[1]Wen, Chuan, et al. "Keyframe-focused visual imitation learning." *arXiv preprint arXiv:2106.06452.* 2021.

---

### Meta-Review · Area_Chair_u8Aj · 2024-12-21

**Metareview:**

This paper addresses the problem of causal confusion in imitation learning from raw sensory data. The authors identify three factors contributing to causal confusion and propose techniques to mitigate these effects. Experiments conducted in a simulated driving domain demonstrate the effectiveness of the proposed method, with comparisons to several baselines.

The reviewers acknowledge the significance of the problem setting and the novelty of the proposed method. However, reviewers express concerns regarding the technical and conceptual contributions of the work. Specifically, the reviewers highlight the absence of strong theoretical guarantees or empirical evidence supporting the claim that the learned policy is truly causal. They also suggest that the paper's presentation could be improved and that additional analyses are needed to enhance the audience's understanding of the contributions.

Overall, while the paper addresses an important problem and proposes interesting ideas. However, it can be further improved in terms of analysis, experiments, and clarity of presentation. The authors are encouraged to address these aspects in future revisions to strengthen the impact of their work.

**Additional Comments On Reviewer Discussion:**

Authors added additional experiments during rebuttal period to address reviewers' concerns.

---

### Decision · Program_Chairs · 2025-01-22

Reject